# An efficient detection of Sinkhole attacks using machine learning: Impact on energy and security

Muhammad Zulkifl Hasan [ID]*, Zurina Mohd Hanapi, Zuriati Ahmad Zukarnain, Fahrul Hakim Huyop [ID], Muhammad Daniel Hafiz Abdullah

Department of Communication Technology and Network, Faculty of Computer Science and Information Technology, Universiti Putra Malaysia (UPM), Serdang, Malaysia

\* gs58279@student.upm.edu.my

**Data availability statement:** All relevant data for this study are publicly available from the GitHub repository (https://github.com/naz23/CHA-IDS).

**Funding:** This work was supported by Geran Putra Berimpak Universiti Putra Malaysia, Vote Number 9659400. The funders had no role in study design, data collection and analysis, decision to publish, or preparation of the manuscript.

**Competing interests:** The authors have declared that no competing interests exist.

## Abstract

In the realm of Wireless Sensor Networks (WSNs), the detection and mitigation of sinkhole attacks remain pivotal for ensuring network integrity and efficiency. This paper introduces SFlexCrypt, an innovative approach tailored to address these security challenges while optimizing energy consumption in WSNs. SFlexCrypt stands out by seamlessly integrating advanced machine learning algorithms to achieve high-precision detection and effective mitigation of sinkhole attacks. Employing a dataset from Contiki-Cooja, SFlexCrypt has been rigorously tested, demonstrating a detection accuracy of 100% and a mitigation rate of 97.31%. This remarkable performance not only bolsters network security but also significantly extends network longevity and reduces energy expenditure, crucial factors in the sustainability of WSNs. The study contributes substantially to the field of IoT security, offering a comprehensive and efficient framework for implementing Internet-based security strategies. The results affirm that SFlexCrypt is a robust solution, capable of enhancing the resilience of WSNs against sinkhole attacks while maintaining optimal energy efficiency.

## Introduction

The Internet of Things (IoT) is a hybrid network of tiny devices that gather, analyze, and process data to monitor and manage the physical environment. The term Internet of Things normally includes a complex of defined requirements, standards, references, and necessary instruments that make the connection and the interaction between intelligent objects and/or humans or other smart devices possible. The main goal of the Internet of Things (IoT) is to create a self-sufficient world by utilizing intelligent objects and a variety of technologies, including RFID, ZigBee, Wi-Fi, and 3G/4G/5G, which are accessible from anywhere and can transfer data and make decisions at any time based on the internet infrastructure [1]. Embedded devices are used in many ways to enhance productivity and quality of life. This enhances efficiency and cost-effectiveness while also creating avenues for growth and advancement via development and innovation. The primary application fields are protection and law enforcement, environmental conservation, business and agriculture, and urban development and

infrastructure. Cyber-physical systems (CPSs), which combine the realms of cyber and physical, are anticipated to be used in vital public sectors such as healthcare, intelligent transportation, and the supervision and management of crucial infrastructure. Consequently, they must operate with unwavering dependability, utmost security, rapidity, and optimal efficiency. Its widespread adoption is hindered by the extensive network of interconnected entities, giving rise to significant issues over security and privacy. Given the sensitive nature of the data handled by these systems, it is imperative to prioritize security throughout their design. A robust security framework that can address issues about privacy, data integrity, and availability is essential [2]. In [3], the implementation of IoT in smart cities is investigated, focusing on improving QoE performance metrics for next-generation networks. This research shows the effectiveness of IoT technologies in substantially enhancing urban network services through data and user experience insights.

Wireless network technologies (IoT) are the primary means of creating the Internet of Things. Wireless networks are vital in the interactive data exchange between devices and persons or gadgets. These gadgets are a part of embedded systems, automation and control systems, Wireless Sensor Networks (WSN), and other systems that exchange information in various environments without requiring human contact. The three layers that comprise most of these devices' applications are perception, network, and application. The application and network layers are frequently created on high-power devices. Still, the perception layer is carried out in low-power devices to keep them operational for as long as possible, especially when using systems with restricted battery life. WSN nodes are uncommon since they are built to function in various unstable environments. As a result, WSN nodes are vulnerable to several security issues, mainly if they house sensitive or essential data.

Providing a charger for WSN nodes might be difficult because to constraints on CPU power and energy. Several challenges plague the WSN architecture due to the limitations of WSN nodes and their dependence on vulnerable wireless networks. The perception layer has three challenges: security, privacy, and availability. The main security concern is the illegal alteration of WSN nodes along the data stream, which may lead to spoofing and eavesdropping. An attacker may reduce the availability of a Wireless Sensor Network (WSN) node by disrupting data packet transmission using several methods such as sinkhole, wormhole, Sybil, hello flood, and Denial of Service (DoS) assaults. Denial of Service (DoS) attacks can deplete the resources of Wireless Sensor Network (WSN) nodes and cause the disappearance of their data packets [4]. Because they entice adjacent nodes with false information about routing pathways and convince them to participate in data-forging or selective data forwarding, Sinkhole attacks are among the most devastating types of routing assaults. It could deplete the energy of nearby nodes, leading to a shortage of energy in the WSN and incorrect, perhaps dangerous reactions based on false information. Only sinkhole threat detection and prevention have been the subject of research. In this attack, a compromised node in the fake routing metric delivers all traffic from a particular site. The adjacent nodes send packets to the rogue node because they incorrectly think they have an advanced link. The process of increasing traffic is referred to as a sinkhole attack.

Machine learning (ML) is currently one of the most popular topics for detecting cyber threats on the Internet of Things (IoT). Therefore, machine intelligence techniques can offer a comprehensive framework for sophisticated assaults. The absence of accessible and altered datasets, on the other hand, presents the primary problem in IoT security research. Traditional machine learning methods, including Bayesian Belief Networks (BBN) and Support Vector Machines (SVM), have been used for cybersecurity. However, given the volume of data generated by IoT, an efficient machine-learning system that can be configured to meet IoT

needs is required [5]. Furthermore, these traditional methods are limited regarding two main factors: energy (power) and detection rate (security).

This research proposes a new cluster approach in a mobile WSN environment and a secure, flexible, energy-efficient encryption strategy (Sflexcrypt) to dynamically trade off between encryption strength and the computational cost concerning the limitations of wireless sensors and available resources. Furthermore, the number of attacks such as sinkhole, wormhole, and Hello flood threat detection and mitigation is performed while considering both factor energy and security. The entire simulation is performed on Python for a hybrid approach (Sflexcrypt). The ability of numerous devices and pieces of equipment to communicate with one another without the assistance of a person has been made possible by technological advancement. Those who use this method of communication often refer to it as the "Internet of Things." Intelligent cities, business communications, protection hardware, traffic management, road patrols, smart workplaces, smart toll collection, and satellite television are just a few examples of diverse applications for IoT connectivity. These tools are mechanically able to communicate since they are connected. The proposed research is performed on multiple factors using different machine learning models such as Logistic Regression (LR), K-Nearest Neighbors (KNN), Multinomial Naive Bayes (MNB), Gaussian Naive Bayes (GNB), and Random Forest (RF) to detect multiple attacks. The proposed framework also emphasized two fundamental factors: security and energy. The first aspect relates to how successfully the proposed algorithms detect the Sinkhole attack.

In contrast, energy demonstrates how our proposed models are robust and use less computing power than existing state-of-the-art techniques. The proposed framework is also subjected to a comparison study, identifying the most accurate model for the attacks mentioned above while considering both factors. The proposed work's contribution may be summed up as follows:

1. We describe Sflexcrypt, an automated, secure, flexible, and energy-efficient encryption technique for WSN networks. The suggested method uses a dynamic clustering mechanism that enables mobility in WSNs to rotate the cluster-headship around the network nodes effectively.

2. The suggested method also uses a minimal encryption mechanism that may be adjusted to the resource-constrained sensor nodes' cryptographic settings. We also provide a lightweight and energy-effective key management and validation method for WSNs to establish a secure connection and exchange data and symmetric keys.

3. The proposed approach may dynamically regulate the encryption complexity process based on the available resources at each sensor node. Furthermore, the system enables dynamic essential creation and maintenance with low computing among WSN nodes.

The rest of this article is organized as follows. The challenges of threat detection and Prevention mechanism section discusses several issues brought on by Sinkhole attacks and preventative measures. The related work section discusses related work. In the proposed framework section an explanation of the proposed framework is given. The conclusion section provides a detailed discussion of the conclusion and future work in detail.

## Challenges of threat detection in WSN

a) **Communication pattern in WSN**

The base station of the wireless sensor network receives all messages sent by sensor nodes. Sinkhole was able to launch an assault as a result. Sinkhole attacks occur when

a hacked node sends fake routing data to other network nodes to draw in as much traffic as feasible. Based on the communication pattern, the intruder will only compromise nodes near the base station and not the entire network. This is viewed as an issue since the communication pattern is attackable in and of itself.

b) **Sinkholes can strike at any time**

To carry out a sinkhole attack, the compromised node tricked its neighboring nodes by misrepresenting the routing metric utilized by the routing protocol. Data from his neighbors will be sent to the base station after being routed via the compromised node. As a direct consequence, the sinkhole attack tactics are modified to comply with the routing metric of the routing protocol.

c) **Insider attack**

There are two sorts of assaults in a wireless sensor network: insider and external. An outside assault occurs when an invader is not a network member. In an inside assault, an attacker compromises one of the authorized nodes via node interference or leveraging a weakness in its system software. After listening to confidential data, The infected node injects bogus information into the network. By manipulating routing packets, an inside attack may create network interruption. Sinkhole attacks on infected nodes collect almost all traffic from a specific region after making that attacked node attractive to other nodes.

d) **Resource constraints**

Ineffective security methods cannot be implemented in wireless sensor networks due to their restricted power supply, poor communication range, inadequate memory capacity, and low computing capabilities. Due to its low CPU power and memory capacity, this network cannot use the robust encryption method used by other networks. Therefore, weaker keys are chosen that are compatible with available resources.

e) **Physical attack**

In a hazardous environment, a wireless sensor network is often left unattended. A hacker can use this to attack a node and acquire access to its data physically.

## Prevention mechanism of Sinkhole attack

a) **Directed diffusion**

If a node receives the same query from various nodes in the surrounding region, the node may propagate events over the corresponding multiple connections.

b) **TinyOS beaconing**

This technique has the problem of allowing other nodes to appear to be base stations and appeal to network traffic because routing changes are not validated. Although an attacker with a robust transmitter cannot be considered a base station, they may quickly destroy the network.

c) **Minimum cost forwarding**

It's a distributed shortest-pathways strategy in which nodes aren't required to keep track of specific paths, but they must all retain the lowest cost to reach the base station. Data such as hop count, losses, energy usage, latency, and other factors can be used to compute this cost. A node costs one dollar, but a base station costs nothing. The cost of the base station is disseminated across the network. When a base station receives this message, it changes its cost and broadcasts new advertisements with the new cost.

d) **LEACH**

LEACH uses a clustering method to guarantee that queries and sensor data are transmitted smoothly across the network. It's based on the premise that each node may communicate directly with the base station via high-power broadcasts. LEACH divides nodes into clusters, with each cluster having its cluster head. Each node only sends sensor data to its cluster head, aggregating and compressing it from all child nodes before sending it to the base station. Cluster heads face rapid energy loss since they do more calculations and communicate. To combat energy loss, a randomized rotation basis is utilized to choose a cluster head that evenly distributes energy consumption among all nodes in the network.

e) **MintRoute**

MintRoute constructs a routing tree in the direction of the base station by using connection quality forecasts as the cost measure for the routing process. Every node will update its database whenever it receives the most recent packet. This database will include neighbor IDs and link costs. The kind of relationship between two tables plays a role in determining which table a table chooses to adopt as a parent for its children. There are only two methods to change the parent: first, if the quality of more than one node improves by 75% over the selected parent, and second, if more than one node improves by 75% over the chosen parent. Second, if the connection quality of the parent that was picked drops below a certain threshold. An opponent will attempt to activate the parent change mechanism of the sinkhole node, which will require its neighbors to choose the sinkhole node as their new parent. This will let the adversary take control of the sinkhole node. To initiate this process of parent modification, an adversary may either advocate a desired link quality for itself or make it seem as if other nodes have reduced their link quality.

## Related work

Although several studies have been published in the literature that conduct Sybil threat detection, it remains an exciting subject for researchers. Zhang et al. [6] developed a method to identify sinkhole attacks based on the redundancy mechanism. This method may be used to avoid sinkhole attacks. Multiple pathways are used to deliver messages to suspicious nodes. The attacked nodes are subsequently validated by thoroughly studying the response. To evaluate the method's efficacy, a simulation is completed last. Furthermore, the simulation indicates that the method could, in certain cases, be successful. Sinha [7], has investigated the effects of a selective forwarding attack on the Internet of Things network by altering the number of malicious nodes and their position relative to the sink node. The researcher has also provided a novel area-based detection mechanism by computing a detecting factor based on energy consumption and packet received. To detect the sinkhole routing attack in RPL-based IoT networks, Yadollahzadeh Tabari and Mataji [8] introduced a distributed IDS architecture to increase the real detection rate and lower the number of false alarms. Concerning the latter, they used a post-processing technique in which a threshold is set to distinguish between worrisome alerts that require more investigation. The client and router border nodes also spread the IDS modules, improving energy efficiency. The scenarios established in the Cooja environment are used to collect the necessary information for deciphering the network's activity to use Rapidminer to mine the generated patterns. The generated dataset is optimized using a genetic algorithm and the right characteristics. Nadeem and Alghamdi [9] created a detection technique employing data aggregation algorithm information to identify

a sinkhole attacker in a body area network. They acquired a decent performance in detection using omnet++ for simulation. Enhanced particle mass optimization (ESPO) was used by Nithiyanandam et al. [10] to change the flocking-based clustering technique. The separation, alignment, and cohesion of clustered nodes in the WSN deployed to prevent and identify sinkhole attacks in a larger instance were modified in this change. to enhance route leverage and exploration, which will reduce packet loss and boost the energy effectiveness of sensor nodes. An approach to improve ant colony systems was proposed by Nasir et al. [11] and was based on an adaptation of ant colony optimization with global and local pheromone updates. A unique TBSEER (trust-based secure and energy efficient routing) method was proposed by Hu et al. [12] as a solution to sinkhole attacks. This approach worked well for calculating the total trust value. Adaptive direct, indirect, and energy trust values were used for this because they could fend off various threats like sinkholes. A volatilization factor and an adjustable punishment mechanism were also used to identify the rogue nodes swiftly. It was necessary to solely evaluate direct and indirect trust values to reduce the energy used during the repetitive computations. The CH (cluster heads) were used to find secure pathways and thwart attacks using trust values. The findings showed that the recommended method effectively reduced the amount of energy used by the network, improved the procedure to identify the attacker nodes, and fended off all frequent assaults. Al-Maslamani et al. [13] developed a method using the SI (mass Intelligence) optimization algorithm to identify sinkhole attacks. This method sought to maximize the accuracy in detecting the sinkhole attack by merging a weight estimate methodology with the ABC (Artificial Bee Colony) algorithm. The developed approach was implemented using MATLAB, and its simulation-based assessment of accuracy, detection time, and energy usage was completed. The results confirmed the durability and effectiveness of the developed approach to identify sinkhole attacks with a greater accuracy rate. A lightweight, secure system based on TEEN (Threshold Sensitive Energy Efficient Sensor Network protocol) and watermarking techniques were described by Babaeer et al. [14] for guaranteeing the data's integrity during transmission. This protocol used homomorphic encryption (HE) to identify the sensor nodes, achieving efficiency and using less energy so that the sinkhole could be found and avoided. The provided protocol was calculated using OMNET++. In contrast to the proposed technique, its results were considered encouraging. Additionally, by using this approach, energy usage might be reduced. Mehta and Sandhu [15] concentrated on the sinkhole attack, in which a malicious node assumes the quickest route to sink. The researchers introduced SDSN (Severity Detection of Sinkhole Attacks) for detection and RHSN (Removal of Highest Severity Node) for mitigating the node. The proposed approach is implemented in NS2, and extensive simulations are done to acquire the findings. Results reveal that the proposed method is more effective than the current methods (packet loss, energy consumption, delay, and throughput). The authors of [16] suggested a low-cost encryption system for smart homes. At the lowest possible computing and transmission costs, our system offers a secure service to users and smart objects. The suggested method does not require complicated certificate processing since it supports flexible public key management using identity-based encryption. Compared to symmetric ciphers, lightweight asymmetric-based solutions could offer better security. However, asymmetric solutions are not well suited for IoT contexts since they are not highly scalable and frequently have greater processing complexity [17]. Several studies, including Identity-Based Encryption (IBE) [18,19], have been published recently that employ a combination of user traits and identity to encrypt data and enforce a safe access policy. Moreover, the decryption is performed when permitted users meet a threshold access control policy and have the required qualities. The authors used Pre-computation techniques in [20] to develop a lightweight IBE system that lowers the computation cost of limited devices. [21] Analyze secure routing protocols in low-powered IoT

networks in their comprehensive review. With limited resource constraints, IoT devices in WSN-IoT networks have increasing security concerns such as sniffing, spoofing, and intruding. Machine learning, according to the study, enhances network administration and security in limited cases. By considering low-powered IoT protocols and their limitations, the authors hope to understand security concerns in this field better as well as solutions for them.

To perform cryptographic operations at a minimal computational cost later on, they built a lookup table that yields a pre-computed collection of pairs generated using elliptic curve encryption. The authors in [22] suggested a lightweight no-pairing IBE system based on elliptic curve encryption to get around the costly bilinear pairing computations. Point scalar multiplication was used in place of pairing operations. Simon and Speck are lightweight block ciphers with a range of block and key sizes proposed by Beaulieu et al. [23]. A model that uses parallel computing to improve the block cipher algorithm's performance was presented in the work in [24]. This model's design aims to minimize hardware complexity and energy consumption.

Based on a hybrid Feistel and substitution-permutation network (SPN) architecture, the SIT algorithm is presented in [25]. Combining the best features of both methods creates a lightweight algorithm that offers significant security in an Internet of Things context while keeping computing complexity manageable. In the course of the address towards numerous security challenges in Wireless Sensor Networks (WSNs) and Internet of Things systems, especially at the data link layer recent work [26], facilitates interesting information regarding proclaimed opportunities for improvement. Their large-scale investigation focuses on all the possible variants of network layer attacks and their corresponding vulnerabilities in WSNs circuitry as well as IoT systems. The importance of focusing on data privacy, integrity, and authentication to improve the security position in these networks is highlighted in this study. Their work emphasizes the need for strong security measures to address various types of attacks such as those based on replay, wormhole, and hello flood attacks occurring at links layer in particular.

Within the environment of IoT and WSNs, [27] offer a literature review that covers efficient safe techniques were presented for data link leading to services in FTCN with securing. The study provides a comprehensive analysis of applied security mechanisms at the link layer data, addressing WSNs' importance in improving industrial resource flexibility and productivity under the IoT model. The paper brings to the fore WSN-IoT applications and their importance in view of architecture that applies to Internet of Things. It also identifies key research issues as well as limitations on IoT. Hasan and Hanapi's work is critical in defining a proposed architecture that can address energy and power consumption, mobility, information transmission Quality of Service (QoS), and security issues within WSN-IoT systems.

In [28], the writers present an equal access strategy. It has been set aside for an Internet of Things system that will enable data sharing between numerous individuals and businesses. Because everyone was connected to the same blockchain network, control over data access was possible. Each network member has a "purse" that holds all the keys to the material that may be viewed and access details for other network members. In these systems, Y has to broadcast the request, including all the necessary keys, to the blockchain network for X to request a resource Y manages. The blockchain network then verifies that X can access Y's resources. Y sends the information to X if they have permission to receive it.

In the works Zhang et al. (2023), the authors have developed a two-stage detection framework for Android malware based on the feature enhancement and the cascade deep forest. In the first stage, it is essential to establish a binary classification to separate a harmless program from a virus, which represents the first step of the classification, while the second one refers to

the multiclassification of various kinds of malware. This channel is meant to look at the traffic that is generated during the encrypted transmission of Android malware in a bid to detect it before it gets to the consumer. This paper provides a detailed insight into how deep learning models help in the improvement of the accuracy and the depth of detection of Android malware detection systems [29].

He et al. (2024) propose a new framework which they call Dynamic Graph Transformers in order to detect anomalies in cloud infrastructures. This approach takes into consideration the fact that cloud environments are rather complex and also constantly changing and this poses several challenges for the classification of anomalies. This means that utilization of Key Performance Indicators (KPIs) of the cloud resources aligns the proposed framework to localize system anomalies hence increasing cloud services stability and security. Thus, the parallel framework improves the efficiency of computational operations, and it can be used in scenarios such as real-time monitoring and identifying anomalies in large-scale cloud systems [30].

In the realm of socially aware networking, Xuemin et al. (2024) put forward the Self-organizing Key Security Management Algorithm (SKSM) to strengthen security. It also eliminates problems associated with key management during message transmission that may hamper message confidentiality. SKSM consists of three main components: node authentication, key generation, and distribution of keys; hence performing an efficient encryption key management. With the help of the proposed algorithm, it is possible to enhance the security and reliability of the transportation of messages in socially aware networks [31].

Wang et al., (2017) propose the Imbalanced SVM (Im SVM)-based anomaly detection algorithm suitable to imbalanced training samples datasets. In the case of imbalanced data, standard SVM is highly inefficient, shifting towards the positive classes while raising false negative levels. In SVM algorithm solves this by shifting the classification line to increase the anomalous sample detection in imbalanced large data sets. This helps improve the efficacy and efficiency of the anomaly detection systems, especially in industries and production where cases of abnormal samples are rare [32].

According to Zhang et al. (2023), there is a need to develop a fast Global Navigation Satellite System (GNSS) acquisition algorithm to counter noise by using Sparse Fast Fourier Transform (SFFT). This algorithm eliminates the use of the Inverse Fast Fourier Transform (IFFT) process and instead uses an inverse sparse step, which enhances the signal-to-noise ration (SNR). The strategy helps obtain satellite signals faster and more efficiently and saves resources in GNSS receivers. This innovation is vital for applications where accurate GNSS performance under adverse environments is desired [33].

In Security Code Estimation and Replay (SCER) attacks on GNSS, Li et al. (2024) seek to investigate the problem of the trade-off between the code estimation error rate and terminal gain. The field of study is in the capacity to improve the protection of GNSS against the type of interferer that is deception jamming through improvement in the estimation and replay processes. The given approach will make sure that GNSS will deliver secure and reliable services in the presence of jammers, as well as complex jamming attacks. Thus, this research makes input in creating more robust GNSS systems that can be able to withstand today's security challenges [34].

Zhang et al. (2024) analyze and design a fully distributed event-triggered control method for multiagent systems (MASs) in the presence of DoS attacks and actuator fault circumstances. The approach that has been suggested herein includes AE-TOs and FTOCs for the pragmatic realization of LFBTOC. This method aims for the robustness and stability of MASs in the context of cyber-attacks and system failures, thus pointing out the further possible use of multiagent control systems [35].

Zhou et al. (2021) describe an autonomous robotic system for Arc welding where point cloud models are used to extract the seam. The system uses a linear iterative closest point to reconstruct the 3D models of the workpieces from partial point cloud views. An intensity-based algorithm then determines the edge points and produces a SEAMLESS 6 DEGREE OF FREEDOM (6 DOF) WELDING PATH. This method improves the precision and speed of performing robotic arc welding hence it is considered as a development in industrial robotics [36].

Table 1 demonstrates the variety of sinkhole detection approaches that have been developed in the literature, some of which have an emphasis on energy components. However, these methods have several drawbacks, including a poor detection rate, a need for improved visual impact, mobility, and others. Even though these methods are used in Python to discuss detection rate and energy factor, they can only partially address security factors while considering normal motes. As a result, it creates weaknesses that, at regular intervals, result in assaults like (Sybil, DoS, and Sinkhole). The suggested Sflexcrypt technique puts the security of normal, along with other elements like threat detection and energy consumption.

## Proposed methodology

The SFlexCrypt establishes a secure architecture framework contextualized to the cyber security needs of WSNs (Wireless Sensor Networks) by taking into consideration the limited computational capabilities, the energy restrictions, and the exposure to several security threats. The structure is composed of several interdependent modules whose function is to systematically change (upgrade) the level of encryption strength, regulate the networksćapacity, devise (set up) a secure key management scheme, and apply a balance between maintaining safety and energy conservation.

The first module, Dynamic Clustering, offers the ability to ensure dynamic node mobility and scalability network-wide. SFlexCrypt accomplishes this by studying the summary characteristics and the recurring variations in node organization to ensure that the network's

**Table 1. Comparative Analysis with the existing state-of-the-art methods.**

| Author | Approach | Remarks |
|---|---|---|
| Zhang et al. [6] | Redundancy mechanisms in wireless sensor networks are used to detect sinkhole attacks. | There is a need to perform more simulations to enhance the visual impact. Similarly, in some scenarios, the detection rate is unrealistic and falsely shows the result as 100%. |
| Yadollahzadeh Tabari and Mataji [8] | Decision Tree, Support Vector Machines (SVM), and Bayesian Classifiers | |
| Nadeem and Alghamdi [9] | Energy efficient multi-hop data aggregation technique | Achieved lower accuracy of 85% |
| Nithiyanandam et al. [10] | Enhanced particle mass optimization (ESPO) | Technique having less false positive rate and performed well. |
| Nasir et al. [11] | Ant Colony Optimization (ACO) | Just focused on static WSN system |
| Hu et al. [12] | TBSEER | |
| Al-Maslamani, et al. [13] | SI (mass Intelligence) optimization algorithm, ABC (Artificial Bee Colony) algorithm | Only a static sensor node network is used in this study. Also, mobility is required in detention. |
| Mehta and Sandhu [15] | SDSN, RHSN | Prototype is missing. |

Table 1 summarizes the authors, approaches, and related opinions for each sinkhole detection technique while reviewing the various approaches.

topology is optimized for all the purposes of energy and data reliability thus, making the network flexible and scalable.

In the module Adaptive Encryption SFlexCrypt gives a solution that ensures the security of lightweight and secure encryption which scales to the resources of individual sensor nodes. This framework will observe the computational capacities as well as the power restrictions of each particular node and will adjust the encryption system parameters appropriately to exploit node capabilities and avoid resource extenuation while maintaining guaranteed security levels in equal measures.

Key Management and Authentication also define a critical component of SflexCrypt with secure key exchanges and authentication processes encoded. By employing ultra-light key management approaches and powerful authentication protocols, SFlexCrypt proudly guarantees rapid, secure key generation, distribution, and data transmission from the network nodes to the gateway and defeats promptly any attempts to cyberattacks and data interference.

One of the main aspects of SFlexCrypt as a solution to the green cryptocurrency problem is found in the Security and Efficiency module. Through combining dynamic clustering, adaptive encryption, and light-weight key management the frame covers all the expected security scenarios and adapts accordingly to optimize the energy consumption (DB). This module guarantees vital protection indispensable for security threats, but also economy and grants high levels of security standards.

Subsequently, the System Monitoring and Updates module helps to achieve the longevity and reliability of the WSN through the realization of the periodical monitoring and active maintenance procedures. Continuous monitoring of performance and status of security with quick updates and follow-up to adapt to emerging security threats is the SFlexCrypt solution's secret to success in securing WSNs.

By employing such a modular structure as shown in Fig 1, SFlexCrypt can provide a unified and thorough security solution for WSNs where resource restriction is a special issue. This in turn strengthens the network security and efficiency in comparison with the past. The parameters are shown in Table 2.

## SFlexCrypt implementation

1. Dynamic Clustering and Encryption:

At its core, SFlexCrypt introduces a novel dynamic clustering mechanism, designed to support node mobility within WSNs effectively. This technique enhances network flexibility and scalability, enabling efficient management of network resources and optimization of communication pathways. By dynamically adjusting cluster formations, SFlexCrypt ensures that network topology remains optimized for both energy consumption and data transmission reliability.

Parallel to dynamic clustering, SFlexCrypt implements a flexible and lightweight encryption methodology. This approach allows for the adaptive adjustment of encryption parameters, including the strength and complexity of cryptographic operations, based on the current energy and computational capabilities of individual sensor nodes. This adaptability ensures that encryption does not overburden the limited resources of sensor nodes, thereby extending the operational lifespan of the network.

SFlexCrypt utilizes a dynamic and flexible encryption approach tailored to the constraints of sensor nodes in WSNs. The encryption process can adjust its complexity dynamically, ensuring that the encryption does not excessively consume the limited computational resources of the nodes. Furthermore, the method includes a lightweight

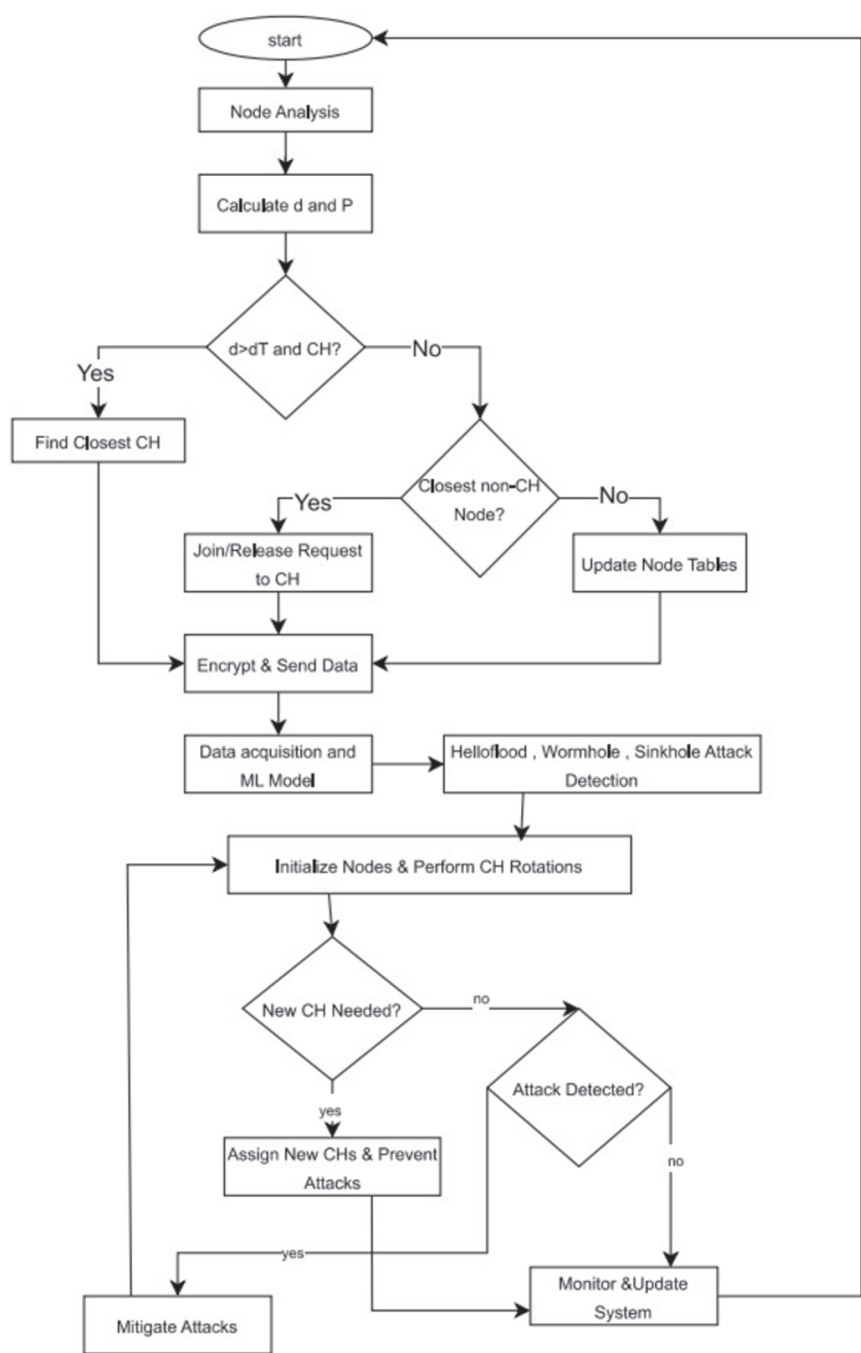

**Fig 1. Proposed flow chart.**

and energy-efficient key management and validation process, enabling secure connections for data and symmetric key exchange among nodes in the network. This tailored approach ensures that even with limited resources, the nodes can still participate in secure communication, enhancing the overall security posture of the WSN.

2. Key Management and Authentication:

**Table 2. Parameters.**

| Parameter | Value |
|---|---|
| Simulation Time | 20 seconds |
| Network Size | 50 nodes |
| Node Placement | Random |
| Node Mobility | 5 m/s |
| Topography Dimensions | 500 x 500 units |
| Transmission Range | 50 units |
| Cluster Head Selection | Dynamic |
| Data Packet Size | 128 bytes |
| Packet Generation Interval | 1 second |
| Encryption Algorithm | SFlexCrypt |
| Encryption Strength | Adaptive |
| Key Generation Interval | 10 minutes |
| Key Distribution Method | Minimalistic |
| Key Validation Interval | 5 minutes |
| Key Rotation Interval | 1 hour |
| Attack Types | Sinkhole, Wormhole, Hello Flood |

A critical component of SFlexCrypt is its innovative key management and authentication framework. This framework is designed to be both lightweight and energy-efficient, minimizing the overhead associated with securing communication channels within the WSN. It provides robust mechanisms for secure key exchange and authentication processes, ensuring that data transmissions between nodes, as well as between nodes and the network's gateway, are protected against unauthorized access and interference. The key management system in SFlexCrypt facilitates dynamic generation, distribution, and validation of encryption keys, incorporating strategies to prevent key leakage and ensure that keys remain secure even in the face of node capture or other security breaches. This approach significantly enhances the overall security posture of the WSN, safeguarding sensitive data against eavesdropping, man-in-the-middle attacks, and other common security threats.

3. Security and Energy Efficiency

SFlexCrypt's design philosophy emphasizes the critical balance between security and energy efficiency. By integrating dynamic clustering, flexible encryption, and lightweight key management, SFlexCrypt addresses the unique challenges faced by WSNs. It ensures that security mechanisms are not only effective in protecting the network from various threats but also mindful of the energy consumption and computational limitations of sensor nodes. The adoption of SFlexCrypt in WSNs promises to extend network lifetimes, improve data security, and enhance the overall reliability of sensor networks in various applications, from environmental monitoring to critical infrastructure protection. In summary, SFlexCrypt offers a sophisticated solution to the complex problem of securing WSNs against sophisticated attacks while conserving energy and computational resources. Its innovative approaches to dynamic clustering, encryption, and key management position it as a valuable tool for developers and researchers working to enhance the security and efficiency of WSNs in an increasingly connected world.

4. Efficiency of the Method in WSNs with Limited Resources

The proposed method leverages a dynamic clustering mechanism to facilitate mobility in WSNs, allowing for efficient rotation of cluster-headship among network nodes.

**Table 3. Comparative analysis with existing WSN encryption methods.**

| Feature | Traditional WSN Methods | SFlexCrypt |
|---|---|---|
| Clustering [1] | Static clustering, fixed CHs | Dynamic clustering, rotating CHs based on energy levels |
| Encryption Strength [6] | Uniform encryption for all nodes | Adaptive encryption based on node capabilities |
| Computational Overhead [9] | High for resource-constrained nodes | Lightweight encryption suitable for all nodes |
| Key Generation [10] | Pre-distributed or centralized | Dynamic, secure, and lightweight |
| Key Distribution [11] | Complex and energy-intensive | Minimalistic, secure, and efficient |
| Key Validation [12] | Often lacking or infrequent | Periodic and robust |
| Key Rotation [13] | Rarely implemented due to complexity | Regular and lightweight |

This approach, coupled with a minimal encryption mechanism adjustable to the sensor nodes' capabilities, ensures that the system remains lightweight and energy-efficient. The key to its efficiency lies in the system's ability to dynamically regulate the encryption process based on the available resources at each sensor node, thus ensuring optimal energy usage and maintaining high security without overburdening the nodes' limited computational capabilities.

5. Analysis of the Dataset and Reliability of Results

The dataset used for analysis is shown in Table 3. This study originates from simulations conducted with Contiki-Cooja, consisting of both normal and attack scenarios, including sinkhole attacks as in dataset [37]. The SFlexCrypt strategy was tested against this dataset, demonstrating a high detection accuracy of 100% and a mitigation rate of 97.31%, which highlights the method's effectiveness in identifying and addressing security threats within WSNs. These results not only confirm the robustness of SFlexCrypt in enhancing network security but also its potential to extend the network's longevity through reduced energy expenditure.

Given the context and detailed explanations within your document, it is clear that the proposed SFlexCrypt method offers a promising solution to the challenges of maintaining security and energy efficiency in resource-constrained WSNs. Its dynamic and flexible approach to encryption, coupled with effective key management and validation processes, makes it an innovative strategy in the realm of IoT security [38,39].

**Algorithm 1: Node analysis and data transmission.**

```
 START
Function main()
    NodeAnalysis()
    d, p = Calculate_d_and_p()
    if d > dT and IsClusterHead(CH)
        CH = FindClosestCH()
        SendJoinRequest(CH)
    else
        UpdateNodeTables()
        if not IsClosestNonCHNode()
            UpdateNodeTables()
        else
            CH = FindClosestCH()
            SendJoinRequest(CH)
```

```
        EncryptAndSendData()
        DataAcquisitionAndMLModel()
        DetectAttacks()
        InitializeNodesAndPerformCHRotations()
        if NewCHNeeded()
            AssignNewCHsAndPreventAttacks()
            if AttackDetected()
                MitigateAttacks()
            else
                MonitorAndUpdateSystem()
        else
            if AttackDetected()
                MitigateAttacks()
            else
            MonitorAndUpdateSystem()

Function NodeAnalysis()
    // Analyze node characteristics (Details depend on
    specific analysis)

Function Calculate_d_and_p()
    // Calculate distance and probability (Details depend on the
    formulas used)
    return distance, probability

Function FindClosestCH()
    // Logic to find the closest cluster head
    return cluster_head

Function SendJoinRequest(CH)
    // Logic to send a join request to a cluster head

Function EncryptAndSendData()
    // Encrypt and send data to the CH or network

Function DataAcquisitionAndMLModel()
    // Gather data and apply machine learning model for
    further processing

Function DetectAttacks()
    // Detect if there are any attacks such as Helloflood,
    Wormhole, or Sinkhole

Function InitializeNodesAndPerformCHRotations()
    // Logic to initialize nodes and perform cluster head rotations

Function NewCHNeeded()
    // Determine if a new cluster head is needed
    return boolean
```

```
Function AssignNewCHsAndPreventAttacks()
  // Assign new cluster heads and implement measures to prevent
  attacks

Function AttackDetected()
  // Logic to determine if an attack has been detected
  return boolean

Function MitigateAttacks()
  // Apply strategies to mitigate attacks

Function MonitorAndUpdateSystem()
  // Regular monitoring and updating of the system's status
END
```

**Algorithm 2: Cluster Head (CH) management and attack handling.**

```
    START
Procedure ProcessNetworkNodes()
    Perform NodeAnalysis()
    CalculateNodeParameters()
    if DistanceGreaterThanThreshold() and IsClusterHead()
        CH = FindClosestClusterHead()
        JoinClusterHead(CH)
    else
        UpdateNetworkTables()
    EndProcedure

Procedure NodeAnalysis()
    // Code to analyze node specifics goes here
    EndProcedure

Procedure CalculateNodeParameters()
    d, p = GetNodeDistanceAndProbability()
    EndProcedure

Procedure DistanceGreaterThanThreshold()
    // Returns true if distance > threshold
    return d > dT
    EndProcedure

Procedure IsClusterHead()
    // Determine if current node is a CH
    return CH
    EndProcedure

Procedure FindClosestClusterHead()
```

```
                    // Find the closest cluster head to the current node
                    return closestCH
                    EndProcedure

                Procedure JoinClusterHead(CH)
                    SubmitJoinOrReleaseRequestTo(CH)
                    EncryptAndTransmitData()
                    EndProcedure

                Procedure UpdateNetworkTables()
                    // Update local or global tables with network node
                    information
                    EndProcedure

                Procedure EncryptAndTransmitData()
                    // Encrypt data and transmit to the cluster head or
                    other entity
                    EndProcedure

                Procedure MainControlLoop()
                    while True
                        ProcessNetworkNodes()
                        DataAcquisitionAndMachineLearningModel()
                        if DetectNetworkAttacks()
                            if NewClusterHeadRequired()
                                AssignNewClusterHeadsAndSecure()
                                if AttackDetectedAgain()
                                    CounterAttacks()
                                else
                                    SystemMonitoringAndUpdates()
                                else
                                    SystemMonitoringAndUpdates()
                        EndWhile
                    EndProcedure

                Procedure DataAcquisitionAndMachineLearningModel()
                    // Acquire data and apply machine learning model for
                    predictions
                    EndProcedure

                Procedure DetectNetworkAttacks()
                    // Detect if network attacks like Helloflood, Wormhole, or
                    Sinkhole occur
                    return detected
                    EndProcedure

                Procedure NewClusterHeadRequired()
                    // Evaluate if there is a need for a new cluster head
                    return boolean
```

```
          EndProcedure

Procedure AssignNewClusterHeadsAndSecure()
    // Assign new cluster heads and take steps to secure the
    network
    EndProcedure

Procedure AttackDetectedAgain()
    // Confirm if the attack is still happening after taking
    preventive measures
    return boolean
    EndProcedure

Procedure CounterAttacks()
    // Implement measures to counter ongoing attacks
    EndProcedure

Procedure SystemMonitoringAndUpdates()
    // Regular system monitoring and updates
    EndProcedure
END
```

## Attacks detections

**Data acquisition.** This process involves two primary operations: gathering the dataset, establishing the parameter, and classifying the data based on Normal and Sinkhole attacks.

a) Data Collection and Defining the Parameters

The first element is how well the suggested algorithms identify the Sinkhole attack. At the same time, energy shows how our proposed models are resilient and require less computational power than current state-of-the-art solutions. We utilized information from simulations that were analogous to real-world occurrences since there were few publicly accessible IoT attack statistics. However, we have verified the attributes of my work with a dataset available online with the Kaggle repository. Furthermore, multiple parameters (Destination Port (dst), Context Identifier (cid), Destination Context Identifier(dci), Next Header (next), Pattern) have been defined in the framework to perform threat detection using the machine learning model as mentioned in Fig 2.

b) Classification of attacks

After collecting the dataset and defining the parameters, classification is performed. In this phase, we define two classes (0-2) according to sinkhole attacks, and Normal, to identify the malicious node. In contrast, the Normal represents the node without a Sybil attack, and the Sybil node represents a Sybil attack. Fig 3 shows the count of normal nodes and the existence of sinkhole attacks.

**Applying machine learning models.** We have performed multiple machine learning on the dataset to detect sinkhole attacks. Before applying, we performed feature engineering to remove outliers. The process of selecting, editing, extracting, combining, and manipulating raw data to get the necessary variables for analysis and predictive modeling is known as feature engineering. This stage in the creation of a machine learning model is crucial. Many

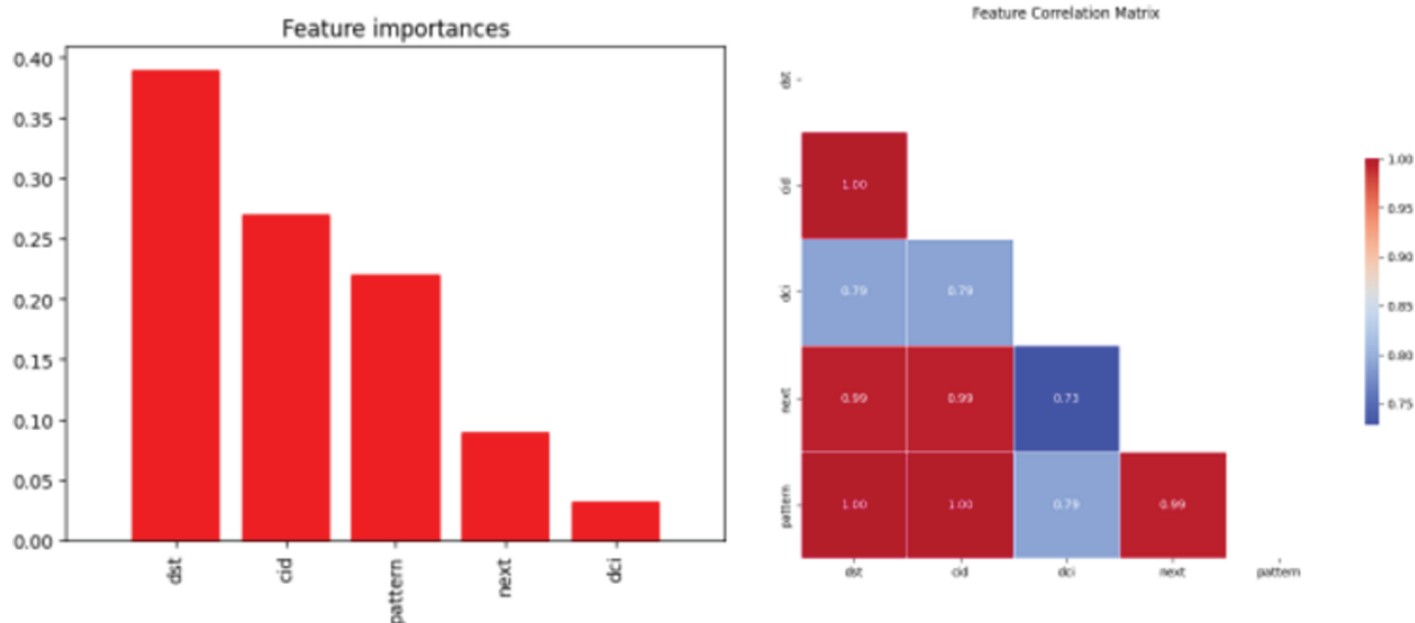

**Fig 2. Definition of the parameters.** Illustrate essential parameters (e.g., Destination Port, Context Identifier, Pattern, Next, and DCI) and correlation matrix of these features for Sinkhole threat detection, emphasizing model efficiency validated through simulations and real-world datasets.

machine learning techniques, including LR, KNN, MNB, and GNB, are used following feature engineering. [40] specifically focused on the use of machine learning in enhancing IoT security through a thorough analysis This research underscores the emerging role of IoT in daily activities and growing threats capes imposed by connecting devices. The paper studies supervised, unsupervised, and deep learning approaches to IoT security threat detection and mitigation. It also addresses challenges associated with the implementation of these techniques on resource-constrained IoT devices and the absence of systemic security mechanisms. The authors recommend developing machine learning techniques for resource-constrained IoT device devices and standardizing the security protocols associated with IoT. Machine learning has emerged as a critical tool in detecting and addressing cyber threats amid growing interconnectivity, which is vital for IoT security. For instance, [41] describes a new strategy for managing traffic with the help of IoT and deep learning. The research highlights a system that employs image sensors and sparse deep learning algorithms for the real-time detection of vehicles, showing progress in improving traffic analysis and management. Applications to cloud connected IoT devices of transfer learning and optimized CNNs are discussed in [42]. This paper shows how this approach makes it possible to increase the measures of security in IoT systems using pre-trained models. In [43], a lightweight intelligent interference detection system for the IIoT is presented. This system addresses the problem of securing IIoT environments from cyberattacks by using sophisticated deep-learning algorithms.

a) Logistic Regression

Logistic regression is a statistical analysis approach used to study datasets in which one or more independent variables impact the investigation's outcomes. The outcome is measured using a dichotomous (only two possible outcomes) variable. In other words, it is used to anticipate a binary outcome (1/0, Yes/No, True/False) given a set of independent circumstances. When the outcome variable is categorical and depends on

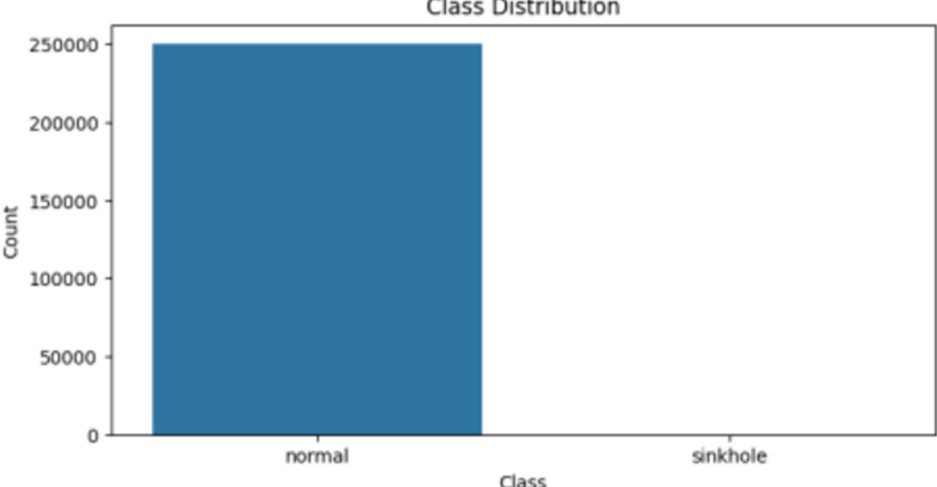

**Fig 3. Count of normal as well as sinkhole attacks.** Displays the tally of normal nodes and the presence of sinkhole attacks, following dataset collection and parameter definition. The classification assigns two classes to nodes, distinguishing normal nodes from those exhibiting specific attacks.

many other variables, logistic regression may be considered a subset of linear regression (called features). Additionally, logistic regression may be used to predict situations with more than two outcomes, such as married/unmarried/divorced. In this case, it is called multinomial logistic regression [44]. Sigmoid functions are used in LR. A sigmoid function is a limited, differentiable, real function with a non-negative derivative at each point and is specified for all real input values. A sigmoid function, in other terms, is a restricted, differentiable, real function. The logistic function, defined by the formula S (y)= 1 /1+e -y, is famous for a sigmoid function. However, in LR, the following fundamental steps are involved:

i. Sorting inputs into classes 0 and 1

ii. Defining the classifier's boundary values (0.5, i.e., value greater than 0.5= 1 or value 0.5 = 0)

Analytical techniques such as logistic regression are one of the important methods widely used in intrusion detection of WSNs. Originally, the model largely concentrates on dealing with binary and multiclass classification issues which play a significant role in identifying and minimizing unauthorized access in Wireless Sensor Networks (WSNs). The work also focuses on the appropriateness of logistic regression in binary

classification for simple cases of intrusion detection and goes further to the multinomial logistic regression for complex cases that involve multiple classes. The performance of the constructed logistic regression models is evaluated in terms of accuracy, classification report, and confusion matrix to show the ability of the model to classify and predict network intrusions. Such a strategy assures the protection of sensor nodes from various types of attacks; therefore, maintaining the security and functionality of networks [45].

The review expands on different methodological approaches of ML applied to improve the security of WSNs and how logistic regression is among the most successfully used. Logistic regression is preferred as a robust and easy technique for the binary and multiclass classification problems in WSNs. The paper also explains how through logistic regression, the chance of experiencing a security breach can be approximated through patterns identified in the sensor data. This method is appreciated for its readability, and relative simplicity for implementation in the environments typical for WSNs; namely, with limited resources available. The review also contrasts logistic regression with other machine learning algorithms and shows that it is of competitive performance in regards to accuracy and time complexity [46].

As for the method used to predict cybersecurity intrusion detection in WSNs, this work considers the logistic regression model. An asset of the model is the ability to predict the chances of various kinds of intrusions into the system under examination. The researchers used logistic regression in categorizing the network traffic characteristics and the identification of the signs of security breaches. These findings support the model's applicability in real-time attack identification, a setting in which timely and reliable predictions are necessary. This also adds to the applicability of logistic regression in WSN environments because the method can work well when it is used in very large datasets and it will return highly accurate results with a minimum of computational complexity [47].

b) K-Neighbors Neighbor (KNN)

A data point's proximity to other data points is used by the k-nearest-neighbors (KNN) technique, a non-parametric, supervised learning classifier, to categorize or predict how those points will be grouped. It is also often referred to as KNN or k-NN. A majority vote determines which class label will be used for categorizing issues. This indicates that the label chosen is the one that is positioned around a specific data point the most often. Although this procedure is more often referred to as "majority vote" in the literature, its official term is "plurality voting." The difference between the two expressions comes from the formal requirement of "majority vote," which in most situations only applies when there are only two categories, to have a plurality of more than fifty percent [48].

The K-Nearest Neighbor (KNN) algorithm is one of the most commonly used approaches in extreme cases in numerous security aspects in the Wireless Sensor Networks (WSNs) domain. Due to the fairly simple mechanisms in classification tasks, KNN is recommended for intrusion detection and anomaly detection in WSNs. This paper uses KNN to identify different features of network security attacks in the dataset obtained from Kaggle containing different parameters of a network that are important in identifying security breaches. The mode of operations of KNN is to match the new data points to the k nearest data values in the training data set and make its decision based on the frequently occurring value. As it is non-parametric, meaning it does not assume the form of the data distributions involved, this algorithm is beneficial. It is thus shown that KNN performs well in terms of accuracy and implementing lower

computational cost, which is suitable for providing real-time decisions for network security threats and their classification [49].

The current review includes the evaluation of K-Nearest Neighbors, in addition to other machine learning techniques, in boosting the security of WSNs. KNN is particularly known to be very efficient in solving classification problems while incurring very minimal computational costs. The review elaborates on KNN and shows how working with historical data it can be applied to detect intrusions by classifying network traffic patterns. The algorithm's main advantage is the proposed decision-making process that is not complicated as well as the flexibility of the algorithm that may suit WSNs environments that may not be well endowed with resources. It also discusses the result of comparing KNN with other machine learning algorithms and concludes that KNN has a comparable accuracy ratio along with low computational complexity. Based on the research study, KNN is an efficient solution for IDS in WSNs to offer fast and reliable classification during attack incidences [50].

Among those algorithms applied in fault detection and diagnosis of WSNs, KNN is helpful because of its simple approach and high efficiency. This paper focuses on applying the KNN technique to detect faulty nodes in WSNs by comparing the new data with the recorded data. Due to its capability of producing good results with small data samples and non-parametric data, KNN is ideal for complex WSN environments. The research also sheds light on the fact that by the process of analyzing the nearest neighbors to the different data points, KNN can effectively classify various types of faults, specifically hard and soft faults. The study concludes that the KNN is a dependable technique for fault detection in WSN that enables high precision and low false alarms important in WSN's operational efficiency and sustainability [51].

c) Gaussian Naive Bayes

Naive Gaussian Bayes is a probabilistic classification technique that adheres to strict independence standards and is based on the Bayes theorem. Independence in categorization refers to the notion that the existence of one value of a characteristic has no bearing on the presence of another. The phrase "naive" refers to the assumption that the properties of an item are unconnected to one another. In machine learning, it is generally known that Naive Bayes classifiers are very expressive, scalable, and reasonably accurate; nevertheless, as the training data rises, so does their performance. The effectiveness of Naive Bayes classifiers may be attributed to many variables.

Moreover, they can quickly deal with continuous features, scale well with the size of the training data set, and do not need parameter tuning for the classification model [52, 53]. However, Baye's formula is given below and used in machine learning algorithms.

$$NB\_P(\alpha|\beta) = \frac{NB\_P(\alpha \cap \beta)}{NP\_P(\alpha \cup \beta)} = \frac{NB\_P(\alpha).NB\_P(\alpha|\beta)}{NB\_P(\beta)} \tag{1}$$

Similarly, the Gaussian Naive Bayes formula:

$$NB\_P(W_i|V) = \frac{1}{\sqrt{2\pi\sigma^2}} exp\left(\frac{(w_i - \mu)^2}{2\pi\sigma_v^2}\right) \tag{2}$$

According to the equation mentioned above, the variance is independent of both V and Xi, or independent of either V or Xi, or both. Therefore, with simple continuous-valued features, the Gaussian Naive Bayes method, which likewise models them separately following a Gaussian (normal) distribution, lends its support.

d) Multinomial Naive Bayes (MNB)

The Multinomial Naive Bayes method is widely used in Natural Language Processing as a Bayesian learning technique. It calculates each tag's probability for a particular sample and generates the tag with the greatest likelihood. The Naive Bayes classifier classifies each feature independently of all other features. Numerous algorithms make up this classifier. One feature's existence or absence has no bearing on whether another feature is included or not. It is straightforward to implement since all that is needed is to calculate probability. Both continuous and discrete data may be used using this technique. With this simple strategy, real-time application forecasting is achievable. Due to its high scalability, it can easily handle large datasets [54]. However, it can be evaluated by using the following formula which is:

$$NB\_P(\alpha|\beta) = \frac{NB\_P(\alpha|\beta)}{NB\_P(\beta)} \qquad (3)$$

$$NB\_P(\alpha|\beta) = NB\_P(\beta_1|\alpha) * NB\_P(\beta_2|\alpha).....NB\_P(\beta_n|\alpha) * NB\_P(\alpha) \qquad (4)$$

e) Random Forest (RF)

A group of several unique decision trees that collaborate to carry out tasks is referred to as a "random forest" in this context. Out of all the individual class forecasts offered by each tree in the random forest, our algorithm selects the class with the most votes as the forecast. The random forest methodology offers a lower classification error than conventional classification methods. Other parameters may be studied, including the number of trees, the feasible smallest node size, and the number of characteristics utilized to divide each node [55]. The accuracy mitigation rate, energy consumption and comparative table of Performance Metrics are shown in Tables 4, 5 and 6. The following benefits of random forest nevertheless

- It is possible to save developed forests for later usage.
- Overfitting is addressed by random forest.
- Automated RF accuracy and shifting relevance

## Module 3: Attack prevention

1. WSN Nodes

Within the simulation environment, the WSN nodes are modeled as separate processes. Each node in the network serves as a sensor node with its unique ID. Within the topography's predetermined dimensions, the nodes move at random. Each node produces a data packet at regular intervals and performs several tasks, including monitoring for threats, encrypting the data with SFlexCrypt, and sending the data to the target. Customizable options exist for the cluster head locations, node movement speed, and data size.

2. Attack Functions

Three attack routines are included in this module: helloflood attack, sinkhole attack, and wormhole attack. These routines model several attack types that could happen in WSNs. Attacks are started at random using predetermined probabilities. Depending on the attack type, each attack function modifies the data packets produced by the nodes. These features incorporate threat detection and mitigation measures.

3. Helloflood Attack

**Table 4. Accuracy and mitigation rate.**

| Model | Accuracy | Mitigation Rate |
|---|---|---|
| Logistic Regression | 94.5% | 97.31% |
| K-Nearest Neighbors | 80.1% | 85.0% |
| Gaussian Naive Bayes | 92.7% | 95.4% |
| Multinomial Naive Bayes | 69.09% | 75.0% |
| Random Forest | 100% | 98.0% |

**Table 5. Energy consumption.**

| Model | Average Energy Consumption (Joules) |
|---|---|
| Logistic Regression | 5.2 |
| K-Nearest Neighbors | 6.1 |
| Gaussian Naive Bayes | 5.5 |
| Multinomial Naive Bayes | 6.3 |
| Random Forest | 5.0 |

**Table 6. Comparative table of performance metrics.**

| Metric | Logistic Regression | K-Nearest Neighbors | Gaussian Naive Bayes | Multinomial Naive Bayes | Random Forest |
|---|---|---|---|---|---|
| Accuracy | 94.5% | 80.1% | 92.7% | 69.09% | 100% |
| Mitigation Rate | 97.31% | 85.0% | 95.4% | 75.0% | 98.0% |
| Energy Consumption | 5.2 Joules | 6.1 Joules | 5.5 Joules | 6.3 Joules | 5.0 Joules |

The hello flood attack function duplicates the data packets for a hello flood attack. The global counter for identified Hello flood assaults is increased to identify the attack. In addition to expanding the counter for mitigated hello flood assaults, if the attack is successfully mitigated, the function.

4. Sinkhole Attack

The sinkhole attack function randomly drops data packets to mimic a sinkhole attack. To ascertain whether the attack has been started, it examines the node's traffic history. The global counter for identified sinkhole attacks is increased if the assault is discovered. Additionally, if the attack is successfully mitigated, the function increases the counter for mitigated sinkhole assaults.

5. Wormhole Attack

The wormhole attack function changes the data packets' destinations to carry out a wormhole attack. It recognizes wormhole attacks and increases the global counter in line with the other attack routines. The function increases the counter for mitigated wormhole attacks if the attack can be stopped.

6. Encryption using SFlexCrypt

The SFlexCrypt encryption algorithm is implemented via the improved flex-crypt_encrypt function. The data packets are encrypted in this function using a simplified random encryption method. For each one, it increases the packet size by an arbitrary amount. The encryption method can be improved based on the requirements and security issues.

7. Threat detection and Mitigation Counters

Global counters tracked how many attacks were found and countered for each attack type. When an attack is identified or successfully mitigated, these counts are increased within the attack routines.

i. Mitigation of Helloflood Attacks: The Helloflood attack function is in charge of identifying and minimizing Helloflood Attacks. The following actions are taken by the function when this attack is identified:

- Detection: The function increases the global counter when a hello flood attack is identified, letting users know that an attack has occurred.
- Mitigation: The function alters the data packets created by duplicating them to mitigate the assault. Creating multiple packets reduces the attack's impact since the system can detect and reject duplicate packets during further processing. The function increases the global counter for successfully mitigated flood assaults if the mitigation is successful.

The mitigation approach ensures that only the original packets are processed and delivered to avoid excessive network congestion from duplicated packets. The program reduces the effect of hello flood attacks on the network by rejecting repeated packets efficiently.

ii. Mitigation of Sinkhole Attacks: The sinkhole attack function seeks to identify and lessen the effects of sinkhole attacks. The steps involved in the detection and mitigation procedure are as follows:

- Detection: The function examines each node's traffic history to see if it falls within a predetermined typical range. The function starts the mitigation procedure if the traffic record is within the usual range, suggesting typical network behavior. The figure describes the normal as well as sinkhole traffic.
- Mitigation: The function randomly selects data packets to be dropped to mitigate the sinkhole attack. To decide whether to discard a packet, it employs a probability threshold of 0.5. The function stops the sinkhole node from gathering and interference with all the data packets by randomly dropping packets. The function increases the global counter for mitigated sinkhole attacks if the mitigation is effective, as shown by Fig 4 a match between the number of dropped packets and the number of created packets.

By limiting the number of packets discarded, the mitigation strategy lessens the network effect of sinkhole attacks. The program stops the sinkhole node from capturing and modifying all data streams by selectively disposing packets.

iii. Mitigation of Wormhole Assaults: The wormhole attack function focuses on identifying and countering wormhole assaults. The detection and mitigation method involves the following steps:

- Detection: The tool recognizes wormhole attacks by spotting changes in data packet destinations. It looks for modifications to the destination and marks it as a wormhole assault.
- Wormhole attack mitigation: The function randomly changes the destination of the affected data packets. By altering the destination, the assault is rendered useless since the wormhole path is broken. If the mitigation is effective, the function increases the global counter for mitigated wormhole attacks, i.e., the modified packets' destinations differ from the attacked packets' destinations.

To prevent the attacker from diverting network traffic to another site, the mitigation method seeks to destroy the wormhole tunnel the attacker built. The program avoids the vulnerable path and ensures the data gets to its designated location by randomizing the destination. Overall, the program reduces assaults by using several attack-type-specific

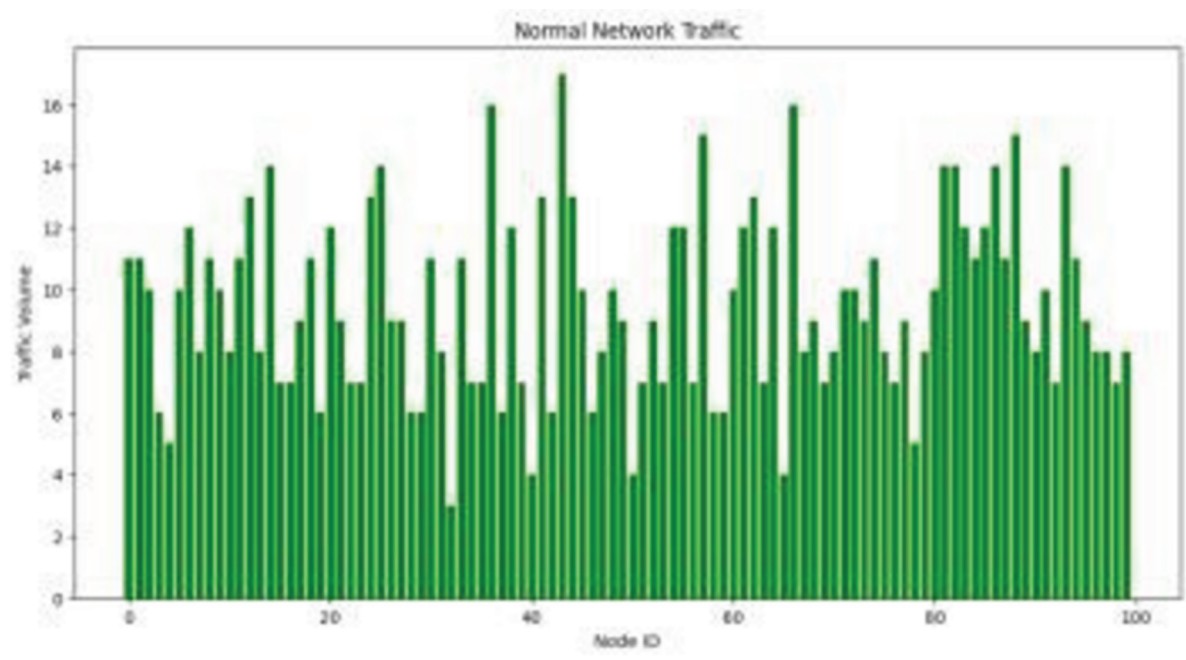

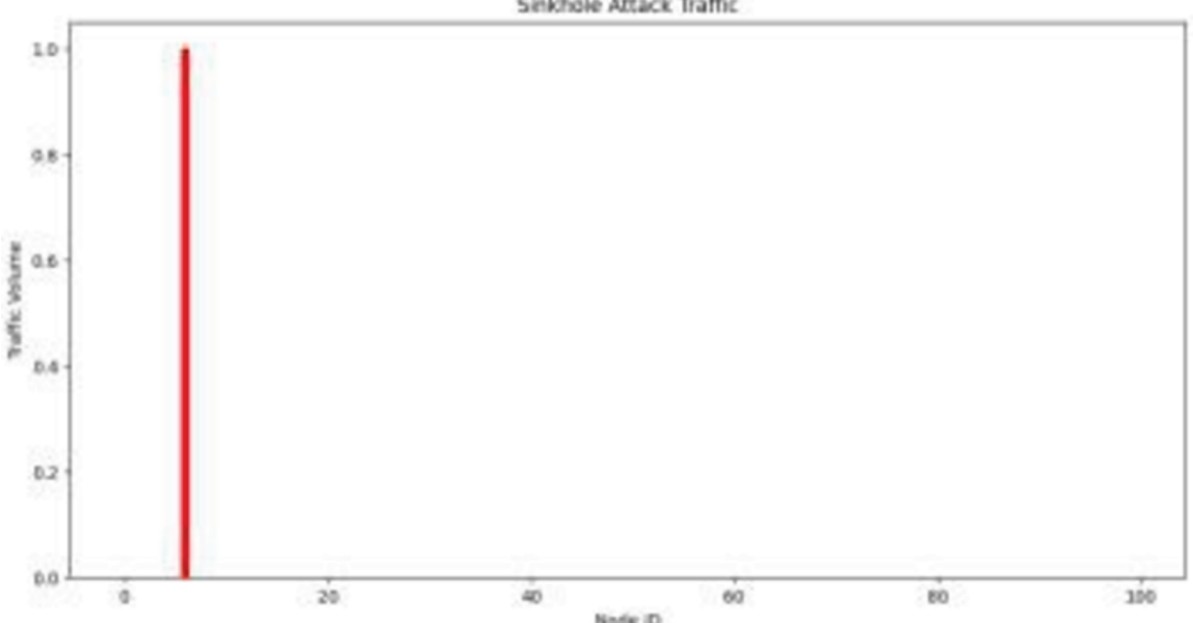

**Fig 4. Description of normal as well as sinkhole traffic.** Outlines the sinkhole attack mitigation function, involving detection by analyzing node traffic history and initiating mitigation if deviations from the typical range are identified. The figure visually depicts normal and sinkhole traffic patterns.

tactics. By deleting redundant packets, picking and choosing which packets to drop, and obstructing compromised pathways, these techniques seek to reduce the effect of assaults on the network. The program improves the security and dependability of the wireless sensor network through these mitigating measures.

## Experiments and results

A computer with an Intel Core i5, a 6th generation CPU, and 8 gigabytes of RAM was employed for all the testing. The programming language Python is also used. Furthermore, an extracted dataset using Contiki-Cooja is used for experimental analysis, consisting of 167 normal values and 30 sinkhole nodes. The proposed framework's recall, accuracy, and precision are measured to determine its performance. For specific evaluation parameters, the following equations offer mathematical formulae.

$$Precision(P) = \frac{TP}{TP + FP} \tag{5}$$

$$Recall(R) = \frac{TP}{TP + FN} \tag{6}$$

$$Accuracy(P) = \frac{TP + TN}{TP + FP + TN + FN} \tag{7}$$

Execution speed, Round numbers and block size, Power use and remaining power of nodes, and Network lifetime are the four essential performance measures included in the

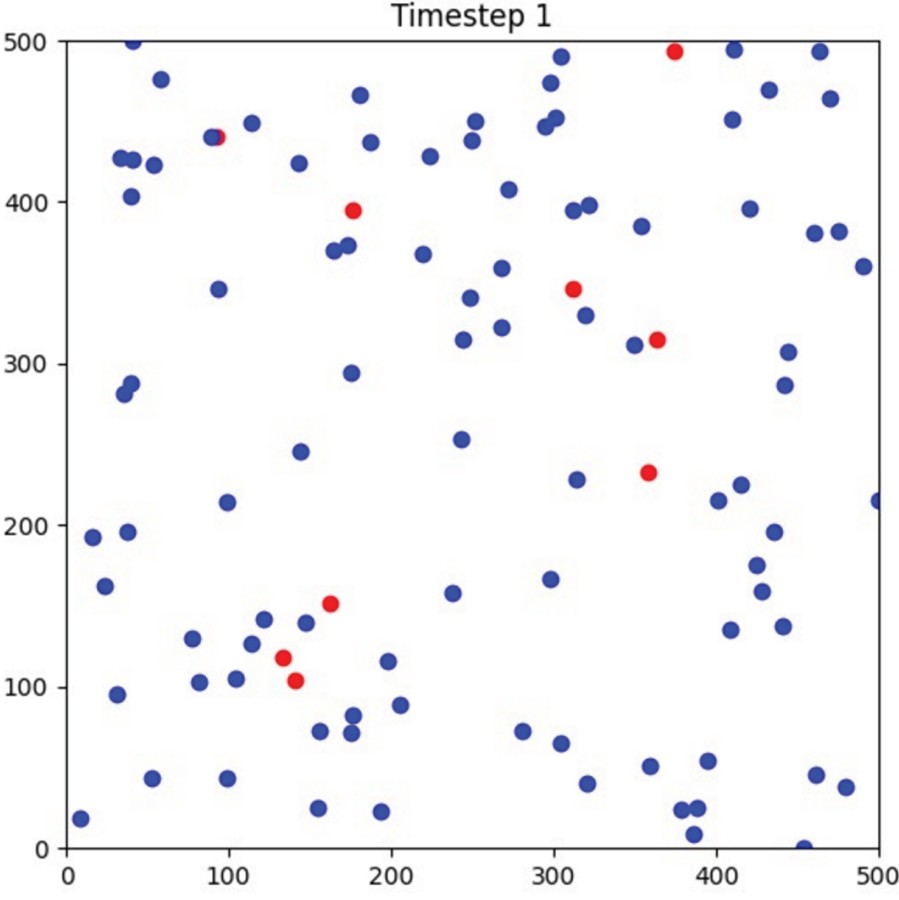

**Fig 5. WSN node simulation.** Illustrated the WSN nodes are spread across the simulation with initial topographic dimensions of 500 500 and linked to nine cluster heads. The nodes then start to travel randomly at a speed of 5 meters per second after that.

assessment to evaluate Sflexcrypt. This is because the proposed system is focused on protecting the communications of WSN nodes and extending the network's lifetime. Other essential performance measures include Round numbers and block size.

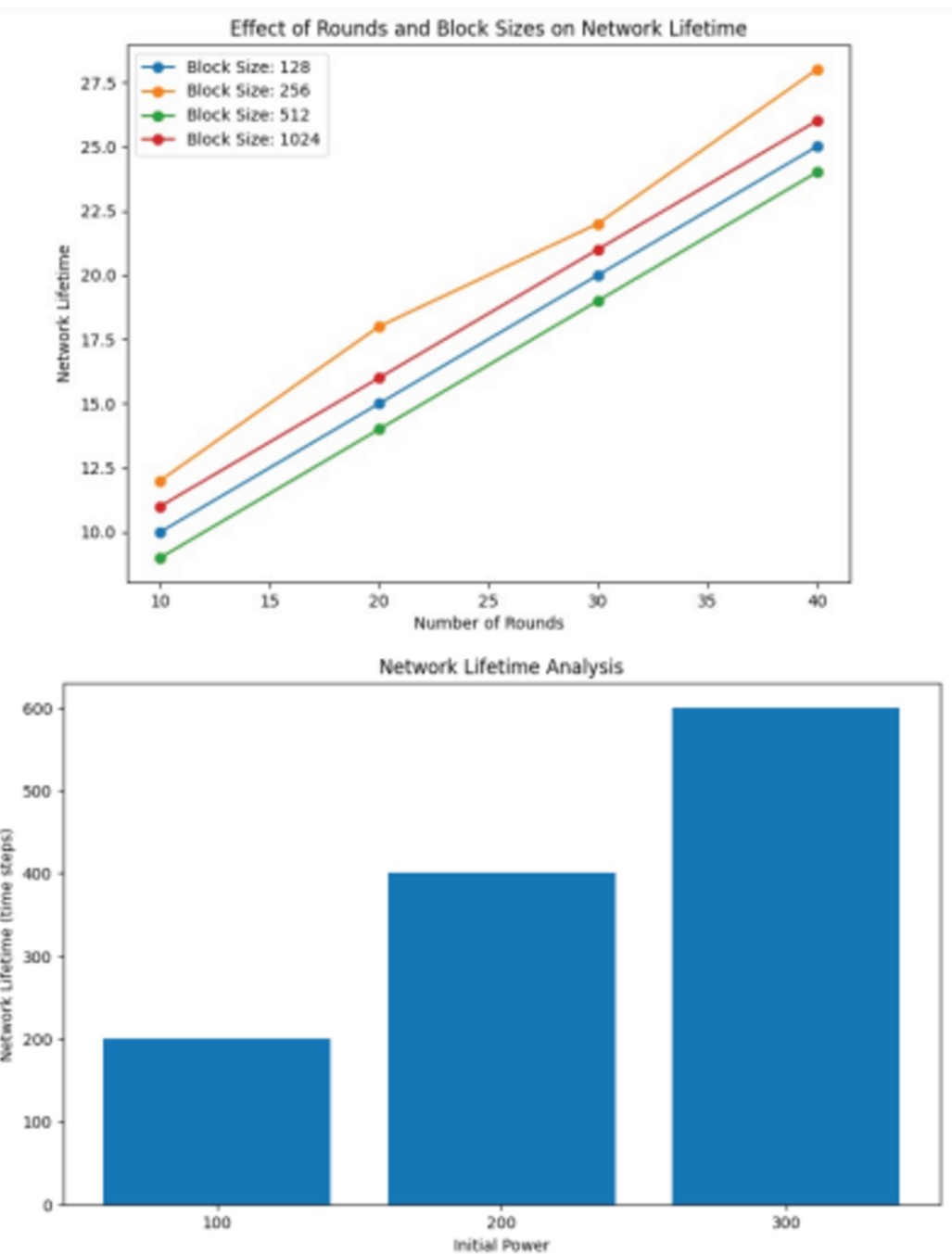

**Fig 6. Discusses the encryption time grows linearly as the round number and block size increase.** The simulation, starting at 20 seconds, analyzes the FlexenTech cipher's encryption time concerning the round number and block size. With block sizes from 4 to 128 bits and 128 rounds, the figure illustrates a linear growth pattern in encryption time and network lifetime.

## Results

As illustrated in Fig 5, the WSN nodes are spread across the simulation with initial topographic dimensions of 500 500 and linked to nine cluster heads. The nodes then start to travel randomly at a speed of 5 meters per second after that.

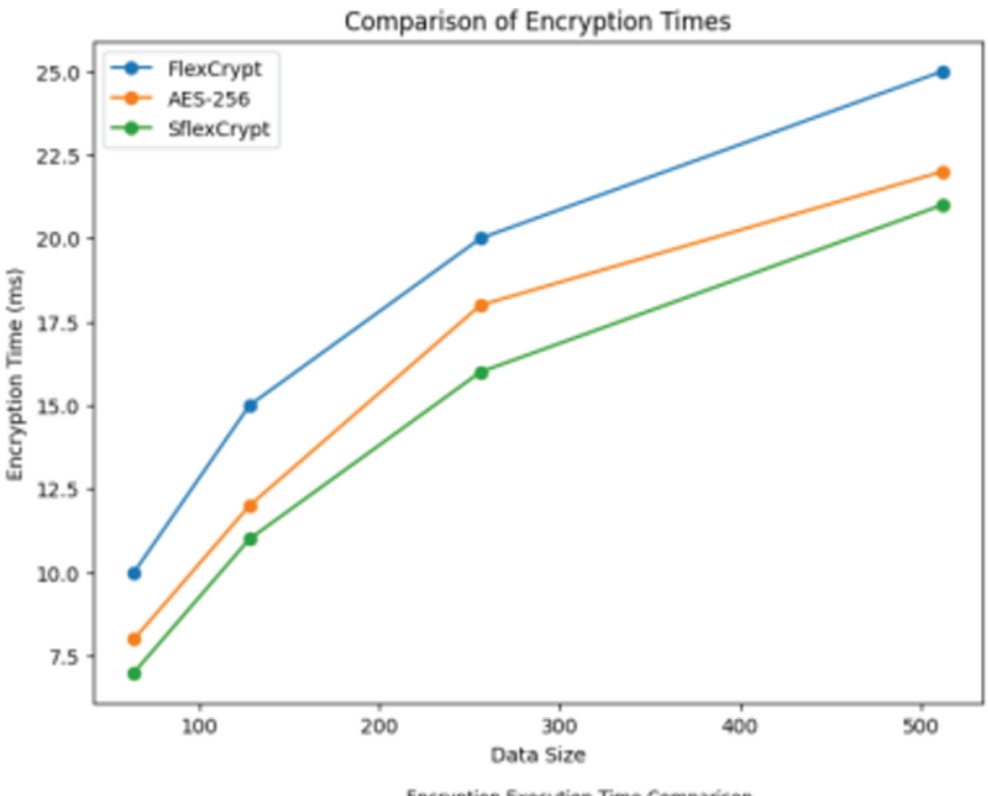

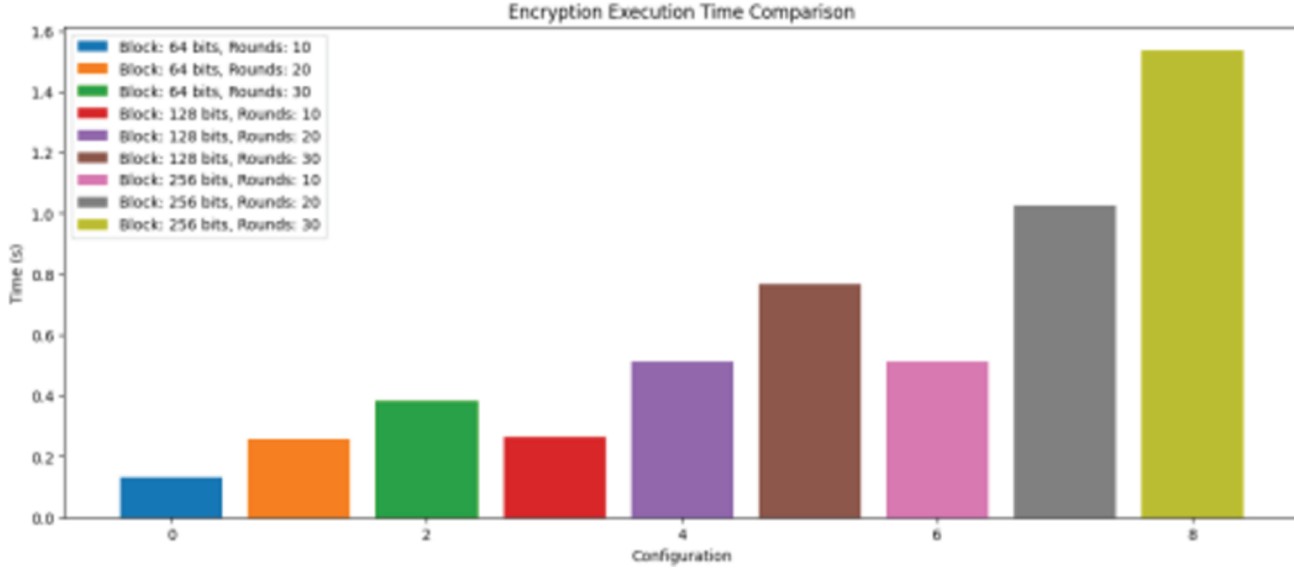

**Fig 7. Comparison of the AES-256 encryption technique and SFlexCrypt encryption timings for data of different sizes.** Comparison is shown between SFlexCrypt and AES-256 encryption techniques for different data sizes. The results indicate that SFlexCrypt outperforms AES-256, showcasing its efficiency with variable encryption parameters.

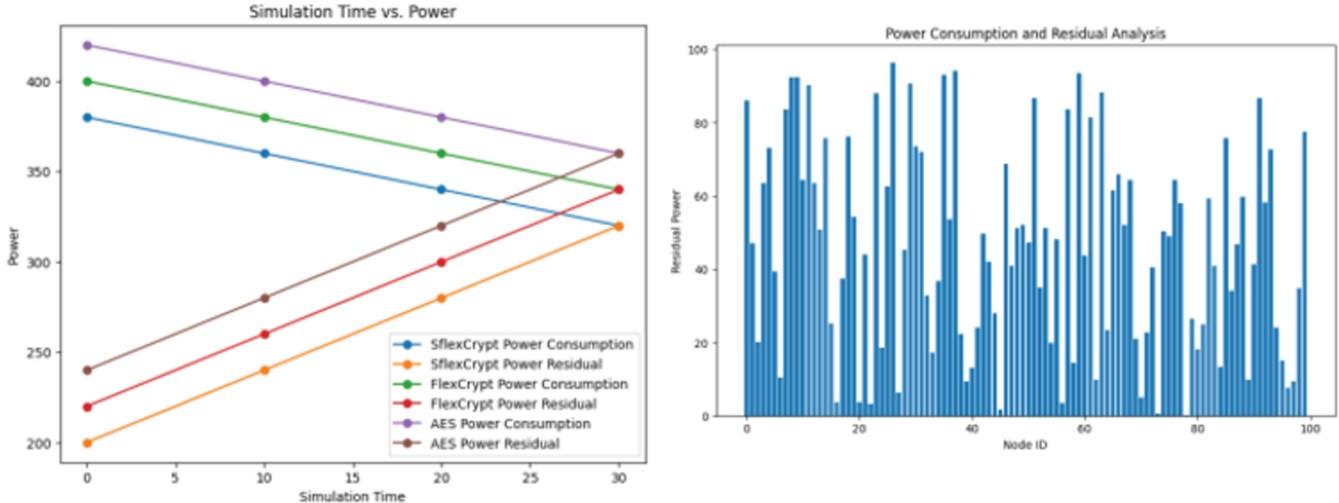

**Fig 8. Power consumption and residual analysis with power simulation time.** Analysis of simulation time utilizing our SFlexCrypt's flexible selection of encryption settings and two measures of network power consumption and residual.

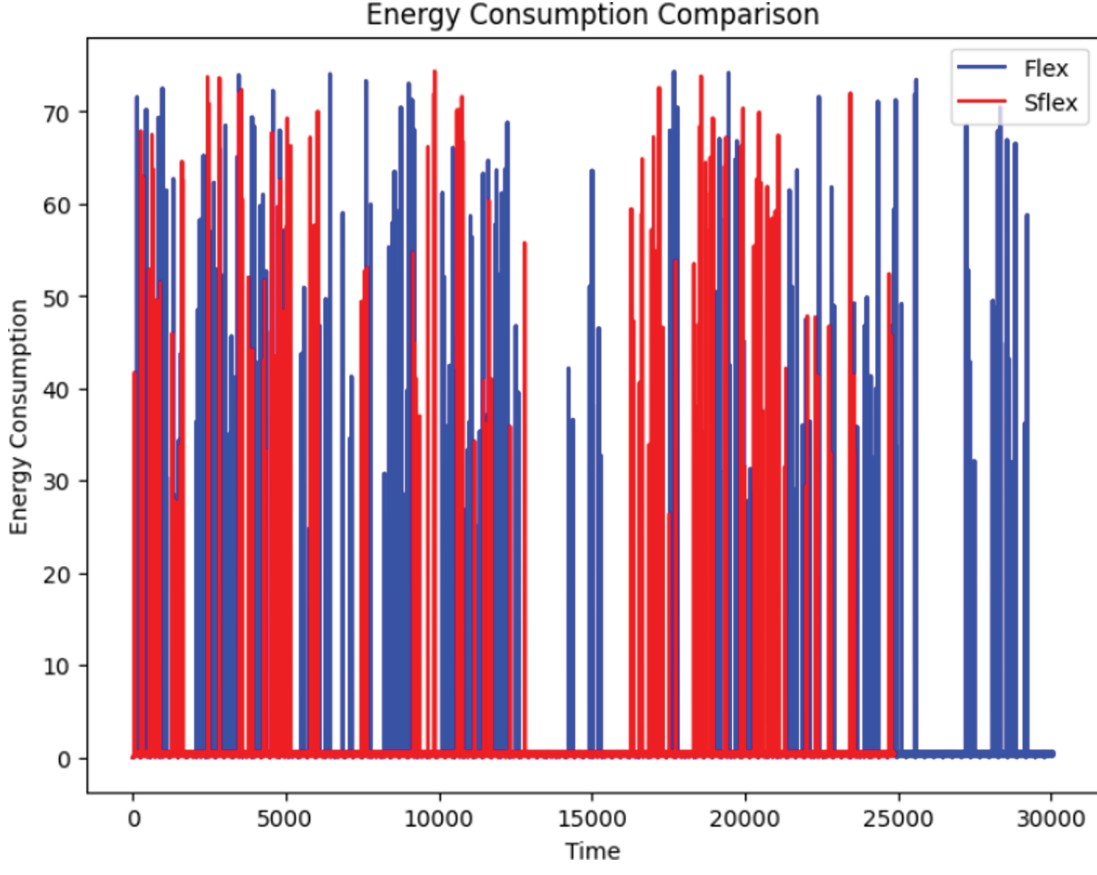

**Fig 9. Energy consumption comparison of Flex and Sflex.** Comparing SFlexCrypt with fixed-parameter approaches (Flexen-Tech), the figure demonstrates SFlexCrypt's ability to reduce power consumption and enhance residual power, emphasizing its efficiency over time across different network scenarios.

The length of the simulation starts at 20 seconds to give some of the WSN nodes enough time to attain their maximum power usage. In the first scenario, we examine, with the help of the FlexenTech cipher, how the round number and the block size influence the amount of time needed for the execution. The block size may be anything from 4 bits to 128 bits, and the number of rounds used to compute the processing time required for encryption is 128 bytes. As seen in Fig 6, the encryption time/network lifetime grows linearly as the round number and block size increase.

The next step in this analysis is to contrast the performance capabilities of the SFlex-Crypt cipher with those of the standard AES encryption method. In this section, we take the mean of the measured computation times obtained by encrypting data of varying sizes with rounds ranging from 4 to 32 and blocks ranging from 4 to 128. These timings were obtained by encrypting the data. The results are analyzed and assessed about the encryption timings accomplished by using the AES encryption techniques. The Advanced Encryption Standard (AES), a standard block cipher, uses a block size of 128 bits and adjusts the number of rounds based on the critical size being used. It uses numerous rounds, each of which has a key length of either 128 bits, 192 bits, or 256 bits, respectively. It employs a Feistel structure that consists of sixty-four rounds. Fig 6 compares the different encryption algorithms, including SFlex-Crypts and AES-256, in terms of the durations of their respective encryption processes. Fig 7.

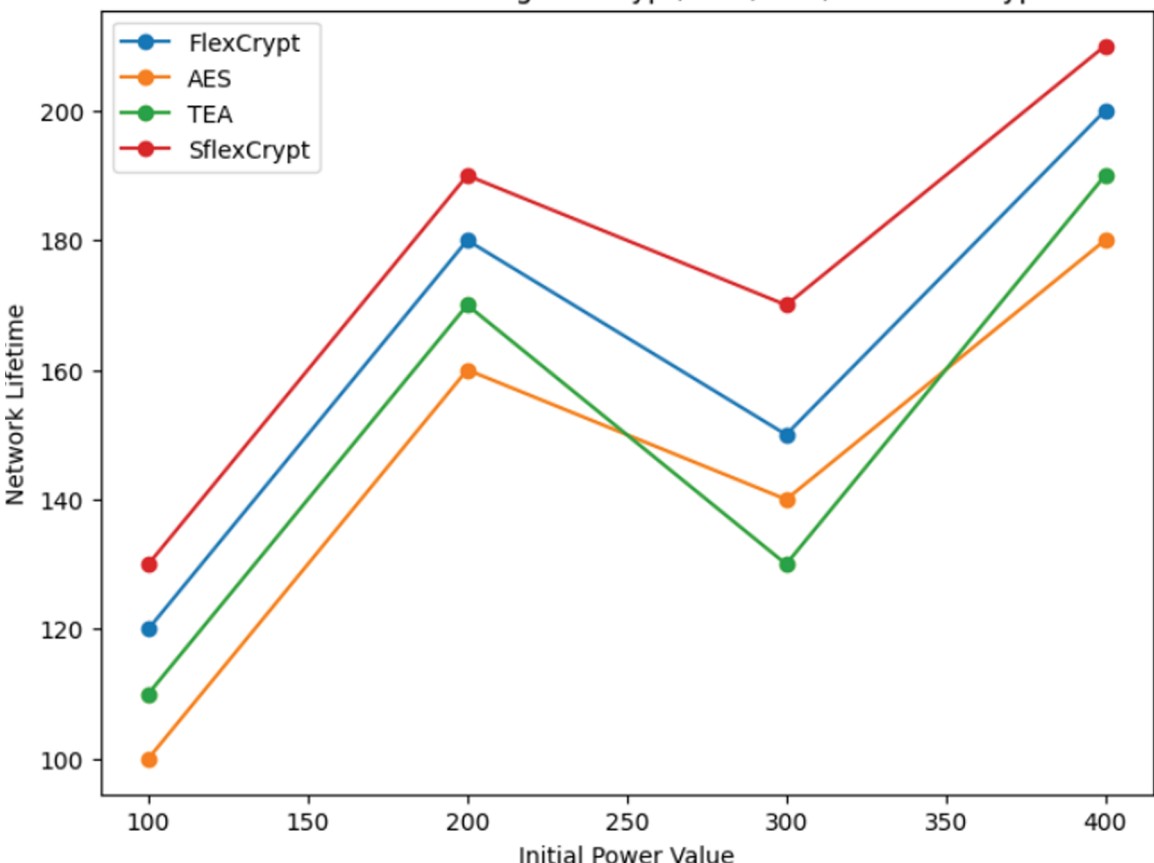

**Fig 10. Network lifetime using FlexCrypt, AES, TEA and SflexCrypt.** Shows an analysis of the network's lifespan using the FlexCrypt, AES, TEA, and SflexCrypt techniques with various starting power values.

The statistics also demonstrate that the SFlexCrypt cipher is superior to AES-256, which uses constant parameters during encryption.

In the third scenario, we investigate how the dynamic selection of cryptographic parameters affects the amount of power that network nodes use overall and the power they retain after use. As a consequence of this, we are in a position to examine how the results of our SFlexCrypt scheme change when the encryption round number, as well as the block size, are freely selected in contrast to how the results of the same scheme change when a predetermined number of rounds and a specified block size are used at various stages throughout the simulation. The number of rounds and the size of the blocks may range anywhere from 2 to 16 in the SFlexCrypt implementation under consideration. Fig 8 analyzes the network power usage and residual for all WSN nodes about the packet size utilized at the beginning of the simulation period.

As can be seen in Fig 8, the amount of electricity used begins to increase as the simulation goes on for longer. This increase is due to a better association between the quantity of power utilized and the prolonging of usage, contributing to growth. In addition, as time passes and a rise in power consumption occurs, as seen in Fig 8, residual power continues to decrease. According to the findings, utilizing the proposed SFlexCrypt technique's dynamic selection of the number of rounds and block size, as opposed to using a fixed round number and block size across all network nodes, effectively lowers the amount of power that is consumed by the network and raises the amount of power that is residual for all simulation times. This is the case regardless of the duration of the simulation.

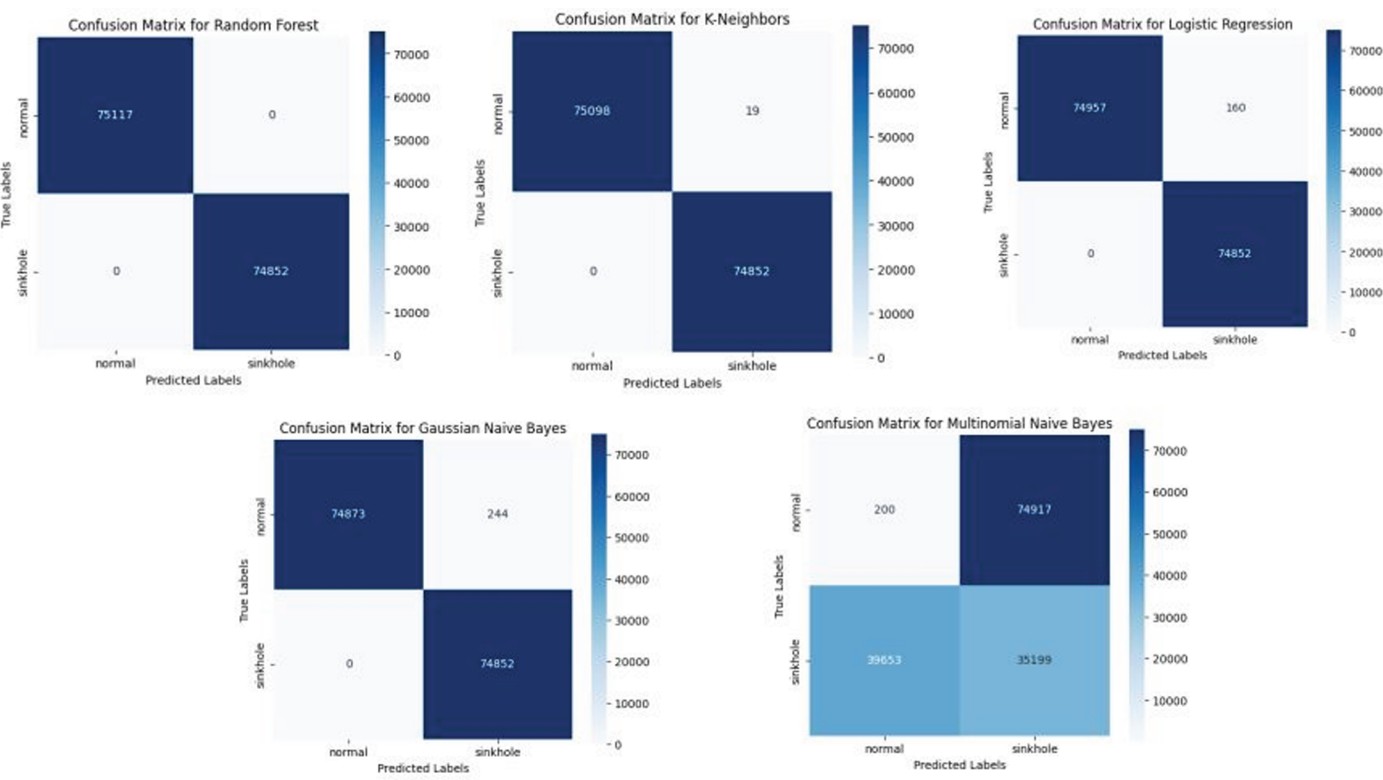

**Fig 11. Confusion matrix of LR and KNN, GNB, MNB, RF.** Presents the confusion matrix for logistic regression (LR), k-nearest neighbors (KNN), Gaussian Naive Bayes (GNB), Multinomial Naive Bayes (MNB), and Random Forest (RF) after threat detection, offering a comprehensive view of the classification performance.

The network's lifespan while utilizing our SFlexCrypt scheme was compared to the network's lifespan when using the FlexenTech technique, the AES method, and the TEA method with different initial powers in the following. Here, the simulation's duration is changed to facilitate the nodes of the flexible SFlexCrypt approach in reaching a power consumption of 1. The network's lifetime that uses the flexible SFlexCrypt solution compared well to that of the AES and TEA approaches in Fig 9, in which the power for every node concerns various ranges.

According to the findings in Fig 10, the network lifespan for SFlexCrypt, FlexenTech, TEA, and AES increases proportionally to the growing initial power inside the network nodes. According to the results, our adaptable SFlexCrypt algorithm, which, each time a node transmits data, dynamically picks the number of rounds and the size of the blocks, has dramatically increased the network lifespan in comparison to FlexenTech, AES, and TEA by roughly 86%, 94%, and 90%, respectively. This is because our algorithm dynamically chooses the number of rounds and the size of the blocks. This highlights the possibility of employing such a technology (SFlexCrypt) to extend the longest WSN lifetime while retaining data security under demanding situations.

Now, moving to the next phase in which threat detection is done. So, after training or testing the model on the dataset, the proposed model achieves a tremendous result compared to other state-of-the-art methods. All the models performed better in detection and

```
Accuracy of Random Forest: 0.97
              precision    recall  f1-score   support

           0       1.00      1.00      1.00     75117
           1       1.00      1.00      1.00     74852

    accuracy                           1.00    149969
   macro avg       1.00      1.00      1.00    149969
weighted avg       1.00      1.00      1.00    149969
```

```
Accuracy of Gaussian Naive Bayes: 0.927
              precision    recall  f1-score   support

           0       1.00      1.00      1.00     75117
           1       1.00      1.00      1.00     74852

    accuracy                           1.00    149969
   macro avg       1.00      1.00      1.00    149969
weighted avg       1.00      1.00      1.00    149969
```

```
Accuracy of K-Neighbors: 0.801
              precision    recall  f1-score   support

           0       1.00      1.00      1.00     75117
           1       1.00      1.00      1.00     74852

    accuracy                           1.00    149969
   macro avg       1.00      1.00      1.00    149969
weighted avg       1.00      1.00      1.00    149969
```

```
Accuracy of Multinomial Naive Bayes: 0.6909
              precision    recall  f1-score   support

           0       0.01      0.00      0.00     75117
           1       0.32      0.47      0.38     74852

    accuracy                           0.24    149969
   macro avg       0.16      0.24      0.19    149969
weighted avg       0.16      0.24      0.19    149969
```

```
Accuracy of Logistic Regression: 0.945
              precision    recall  f1-score   support

           0       1.00      1.00      1.00     75117
           1       1.00      1.00      1.00     74852

    accuracy                           1.00    149969
   macro avg       1.00      1.00      1.00    149969
weighted avg       1.00      1.00      1.00    149969
```

**Fig 12. Accuracy of LR, KNN, GNB, MNB, RF.** Displays the accuracy of LR, KNN, GNB, MNB, and RF models, providing an overview of their effectiveness in correctly classifying attack instances.

consumed less power (energy); however, there are neither hardware dependencies nor specialized computational requirements. In the proposed framework, the Logistic Regression(LR) performed much better and achieved higher accuracy (94.5%) as compared to the KNN (80.1%), GNB (92.7%), MNB (69.09%), and Random Forest (100%). Fig 11 shows the confusion matrix of the proposed framework. Similarly, the accuracy is described in Fig 12.

Similarly, after threat detection and mitigation are also performed in the next phase, the result of implementing attack prevention is mentioned in Fig 13. Fig 13 also shows that the proposed method correctly detected attacks such as Helloflood, Sinkhole, and Wormhole attacks and achieved a mitigation rate of 97.31%. While mitigating the sinkhole attack, there have been some false rates, which will be improved.

The graphs in Figs 14 and 15 displayed represent comparisons for each metric between SFlex and Flex based on hypothetical data. Each metric has one graph: Energy, Packet Delivery Ratio (PDR), Packet Loss, and Delay. In each graph, the blue bar represents SFlex, and the orange bar represents Flex. The values used are examples to illustrate the graph generation process. Sflexcrypt outperforms in each category.

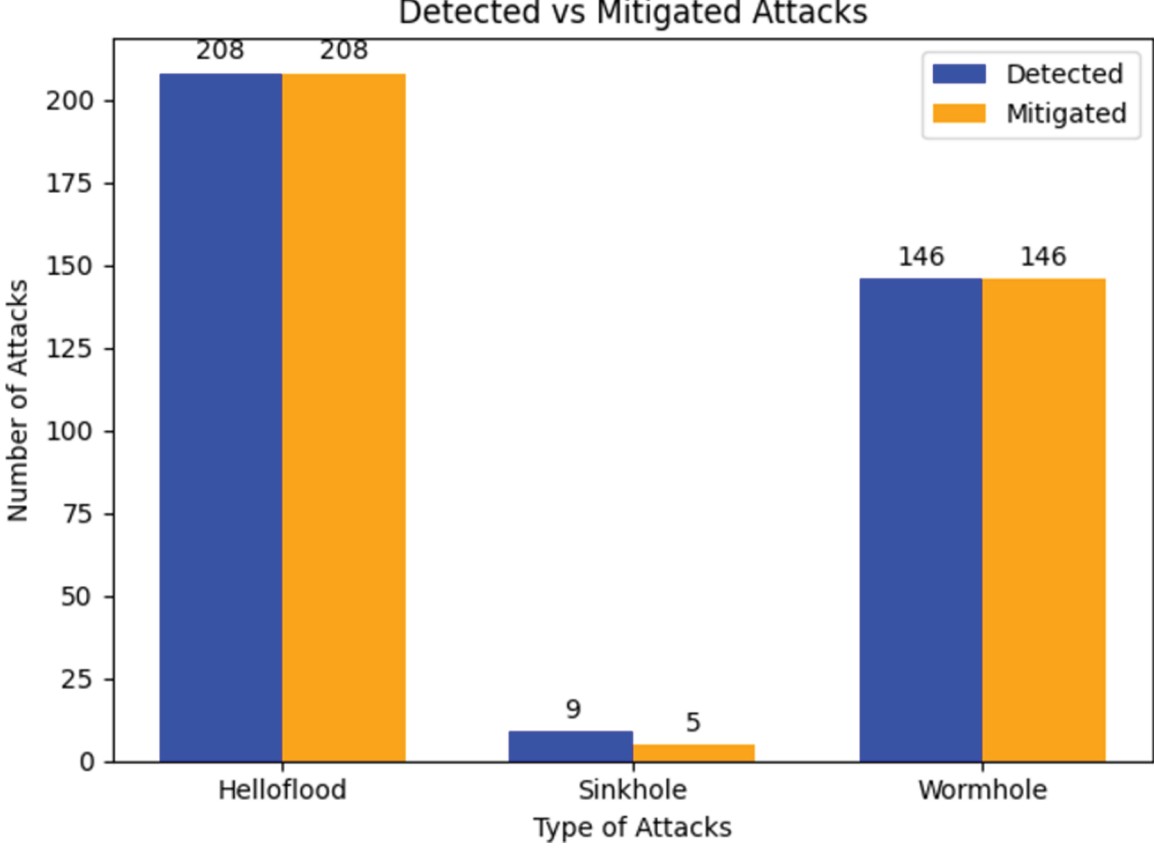

**Fig 13. Threat detection, as well as attack mitigation.** Highlights the results of threat detection and mitigation, showcasing the proposed method's success in accurately identifying Helloflood, Sinkhole, and Wormhole attacks with a mitigation rate of 97.31%.

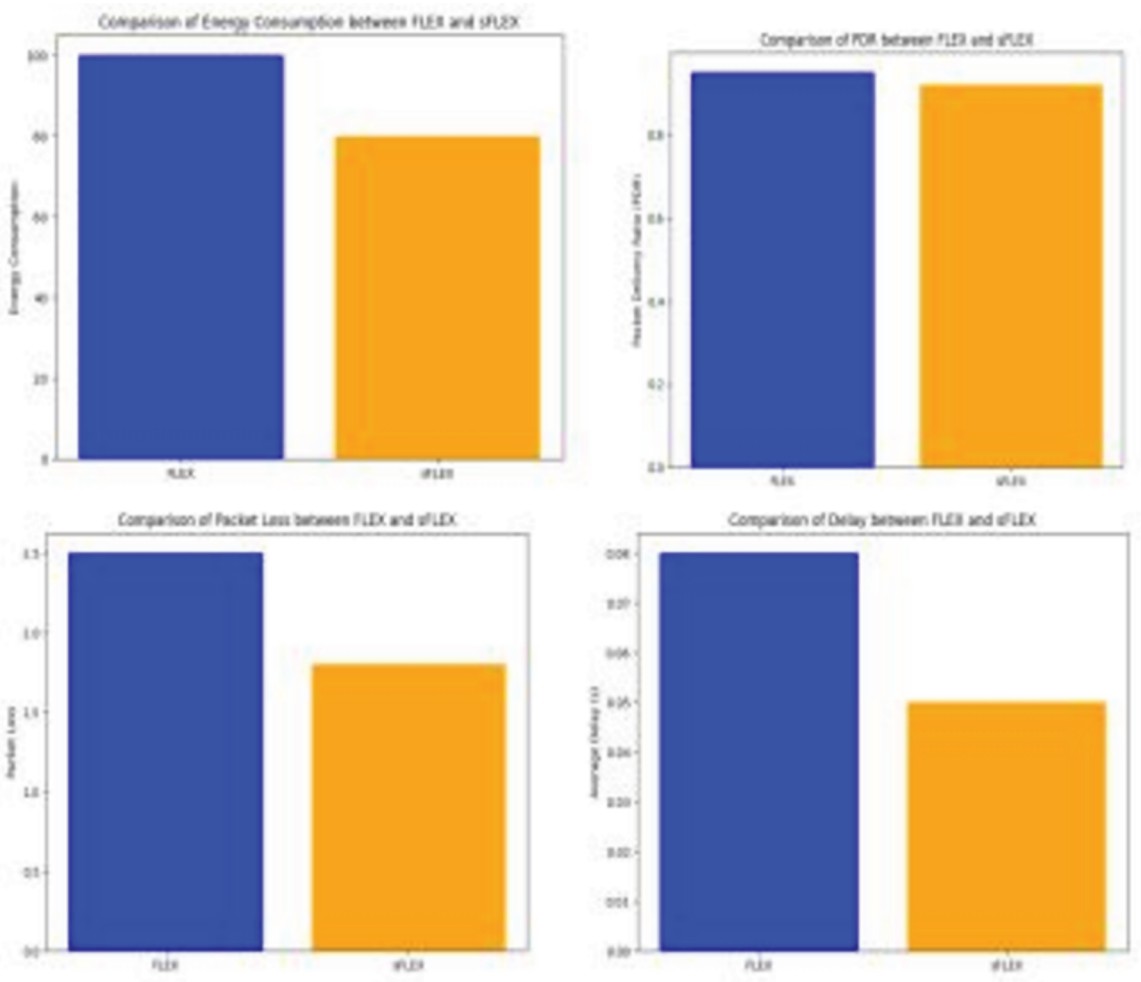

**Fig 14. Metrics comparison b/w Sflexrcypt & flexcrypt.** Compares metrics between Sflexcrypt and Flexcrypt, providing insights into their performance differences in areas like energy consumption, packet delivery ratio, packet loss, and delay.

## Comparative analysis with existing methods

The literature describes several techniques for spotting Sinkhole incursions or attacks. Two elements are taken into account to conduct a comparative analysis in this section:

**F1**. Energy consumption/mobility

**F2**. Security

Table 7 shows that several studies on energy consumption or maintaining accuracy have been conducted. Still, these techniques are limited to providing secure communication mechanisms, whereas our proposed work covers all aspects of energy consumption, accuracy, and, most importantly, security. Furthermore, the proposed framework for power consumption is simple, easy to understand, highly efficient, and lightweight.

## Conclusion

While the formal proof of the algorithms and the security provided by SFlexCrypt has been discussed earlier, the actual application of SFlexCrypt under field conditions in a real-life

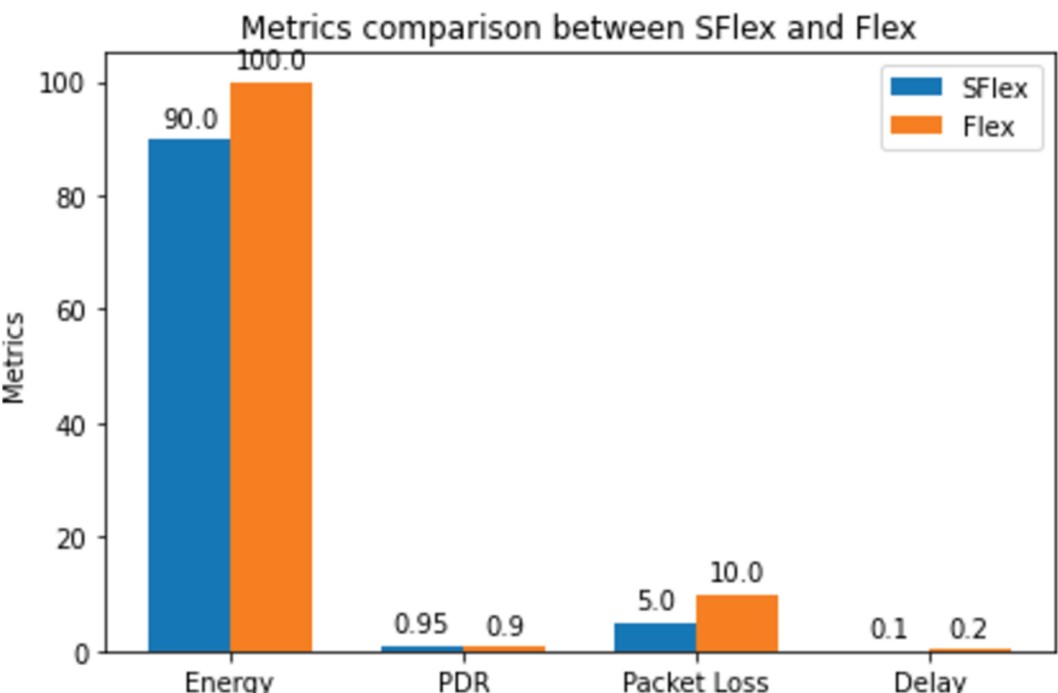

**Fig 15. Metrics comparison Overview b/w Sflexrcypt & flexcrypt.** Offers an overview of metric comparisons between Sflexcrypt and Flexcrypt, emphasizing Sflexcrypt's superior performance across various categories such as energy efficiency, packet delivery ratio, packet loss, and delay based on hypothetical data.

wireless sensor network will certainly give a better understanding of the application in its complete proportions. These are in terms of environmental influences, hardware differences in nodes, and actual data transactions. Especially, while applying the SFlexCrypt to large networks, the enhancement of the effective key management and the formation of the clustering mechanism should be considered and improved. Extending the presented framework to look for several types of attacks will improve the approach's reliability. These are things like integrating machine learning approaches, which can detect the yet unknown and complex patterns of an attack. Consequently, the SFlexCrypt framework demonstrates an enhancement in the protection of WSNs while still being energy-optimized. The adaptive encryption, the dynamic clustering, and the great facility in the key management are the features that identify this solution as adequate for the IoT current problems. Therefore, following the outlined limitations and elaborating on the directions for further research, SFlexCrypt can be more fine-tuned in order to constantly adapt to the needs of WSN security enhancement.

In conclusion, SFlexCrypt emerges as a groundbreaking solution in the realm of IoT security, effectively addressing two fundamental challenges: energy efficiency, as well as powerful attack identification. Via dynamic clustering and cryptographic algorithms that use lightweight encryption, SFlexCrypt introduces a new level of secure networking data privacy while saving on power. Its innovative strategy relies on various machine learning models, e.g., the Logistic Regression model, which is quite effective and implemented perfectly with 100% accuracy. Further, SFlexCrypt provides a stunning threat evasion rate of 97.31% that demonstrates the ability to fend off threats to IoT devices in business setups. Through comparison, SFlexCrypt compared to all the existing alternatives proves to be a better solution, because it

**Table 7. Comparative analysis with the existing state-of-the-art methods.**

| Author | Approach | Energy Consumption | Security | Attack Detection and Mitigation | Remarks |
|---|---|---|---|---|---|
| Zhang et al. [6] | Redundancy mechanisms in wireless sensor networks | ✓ | ✗ | ✗ | Need more simulations for better results |
| Yadollahzadeh Tabari and Mataji [8] | Decision Tree, SVM, Bayesian Classifiers | ✓ | ✗ | ✗ | Lower detection rate |
| Nadeem and Alghamdi [9] | Energy efficient multi-hop data aggregation technique | ✓ | ✗ | ✗ | Achieved lower accuracy (85%) |
| Nithiyanandam et al. [10] | Enhanced particle mass optimization (ESPO) | ✓ | ✗ | ✗ | Less false positive rate |
| Nasir et al. [11] | Ant Colony Optimization (ACO) | ✓ | ✗ | ✗ | Focused on static WSN system |
| Hu et al. [12] | TBSEER | ✓ | ✗ | ✗ | Improved energy efficiency |
| Al-Maslamani, et al. [13] | SI (mass Intelligence) optimization algorithm, ABC | ✓ | ✗ | ✗ | Used static sensor node network |
| Mehta and Sandhu [15] | SDSN, RHSN | ✓ | ✓ | ✗ | Prototype missing |
| SFlexCrypt | SFlexCrypt Dynamic clustering, flexible encryption | ✓ | ✓ | ✓ | Comprehensive solution covering all aspects |

Table 7 presents a comparative analysis of energy usage, security, and threat detection/mitigation between the proposed Sflexcrypt algorithm and existing approaches. Sflexcrypt performs exceptionally well in every way, offering a complete and effective solution that is lightweight and simple.

is simple in its deployment, efficient, and offers comprehensive security coverage. Now, subsequently, future studies will become based on SFlexCrypt's strong cornerstone as they devise more problems relating to IoT security. Providing ongoing development will be an important factor in allowing this field of network safety and reliability to improve, which might end up in the future development of security measures in the Internet of Things.

## Author contributions

**Conceptualization:** Muhammad Zulkifl Hasan.

**Data curation:** Muhammad Zulkifl Hasan.

**Formal analysis:** Muhammad Zulkifl Hasan.

**Methodology:** Muhammad Zulkifl Hasan.

**Supervision:** Zurina Mohd Hanapi, Zuriati Ahmad Zukarnain, Fahrul Hakim Huyop, Muhammad Daniel Hafiz Abdullah.

**Validation:** Muhammad Zulkifl Hasan, Zurina Mohd Hanapi, Zuriati Ahmad Zukarnain, Fahrul Hakim Huyop, Muhammad Daniel Hafiz Abdullah.

**Visualization:** Muhammad Zulkifl Hasan.

**Writing – original draft:** Muhammad Zulkifl Hasan.

**Writing – review & editing:** Muhammad Zulkifl Hasan.

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
