## [Decision Letter · Decision Letter 0]

15 Jul 2024

PONE-D-24-17513An Efficient Detection of Sinkhole Attacks using Machine Learning: Impact on Energy and SecurityPLOS ONE

Dear Dr. Hasan,

Thank you for submitting your manuscript to PLOS ONE. After careful consideration, we feel that it has merit but does not fully meet PLOS ONE’s publication criteria as it currently stands. Therefore, we invite you to submit a revised version of the manuscript that addresses the points raised during the review process.

We look forward to receiving your revised manuscript.

Kind regards,

Gauhar Ali, Ph.D

Academic Editor

PLOS ONE

Journal Requirements:

Additional Editor Comments:

1) In your Data Availability statement, you have not specified where the minimal data set underlying the results described in your manuscript can be found.

2) Please ensure that your manuscript meets PLOS ONE's style requirements.

Reviewers' comments:

Reviewer's Responses to Questions

**Comments to the Author**

1. Is the manuscript technically sound, and do the data support the conclusions?

Reviewer #1: Yes

Reviewer #2: Yes

2. Has the statistical analysis been performed appropriately and rigorously? 

Reviewer #1: Yes

Reviewer #2: Yes

3. Have the authors made all data underlying the findings in their manuscript fully available?

Reviewer #1: Yes

Reviewer #2: No

4. Is the manuscript presented in an intelligible fashion and written in standard English?

Reviewer #1: Yes

Reviewer #2: Yes

5. Review Comments to the Author

Reviewer #1: Reviewer Comments

The paper presents a significant contribution to the field of IoT security with the introduction of SFlexCrypt. Addressing the points mentioned below will enhance the clarity, depth, and impact of the study. I recommend the paper for publication after minor revisions.

Areas for Improvement:

1. Literature Review:

-While the introduction provides a good overview of IoT and WSNs, the literature review could be expanded to include more recent studies and comparative analysis with existing methods.

-Providing a more detailed comparison of SFlexCrypt with other state-of-the-art techniques would strengthen the justification for this study.

2. Technical Clarity:

-Some technical terms and processes are not adequately explained for readers who may not be experts in the field. For instance, the specific machine learning algorithms used (e.g., Logistic Regression, K-Nearest Neighbors) could be briefly described.

-The encryption mechanism and key management process of SFlexCrypt could be elaborated to enhance understanding.

3. Experimental Design:

-The description of the experimental setup, including the hardware and software environment, should be more detailed. Information about the computational resources used and the configuration of the Contiki-Cooja simulator would be helpful.

-Including a flowchart or diagram of the proposed SFlexCrypt framework could provide better visualization of the process.

4. Data Presentation:

-While the results are impressive, the paper would benefit from more detailed presentation of the data. Graphs and tables summarizing the performance metrics (accuracy, mitigation rate, energy consumption) would enhance the clarity of the results section.

-A comparative table showing the performance of different machine learning models used in the study would be useful.

5. Discussion and Conclusion:

-The discussion section should provide a more in-depth analysis of the results. Discussing potential limitations of the study and suggesting areas for future research would be beneficial.

-The conclusion should reiterate the key findings and their implications for the field of IoT security.

Minor Comments:

- Grammar and Style:

-Ensure consistent use of terminology throughout the paper. For instance, terms like "Internet of Things (IoT)" and "Wireless Sensor Networks (WSNs)" should be consistently abbreviated after their first use.

-Some sentences could be rephrased for better readability. For example, the sentence "The expression 'Internet of Things' often refers to a collection of standards, protocols, tools, and technologies needed to link and transport data between smart devices and either other humans or other smart devices" could be simplified.

- Formatting

- Ensure that all references are formatted according to the journal's guidelines.

- Check for any formatting inconsistencies in headings and subheadings.

Reviewer #2: I critically reviewed the " An Efficient Detection of Sinkhole Attacks using Machine Learning: Impact on Energy and Security” paper. We pointed out some weaknesses and provided some recommendations for the improvements. By addressing these weaknesses and implementing the recommendations, the research can significantly improve its impact, applicability, and robustness in detecting latent defects in spindle assembly lines.

1- The paper lacks an extensive literature review on existing requirement catalogs for collaborative software. It would benefit from expanding the literature review section to include more studies on existing catalogs, their methodologies, and findings. I recommend including the following literature:

i. Zhang, X., Wang, J., Xu, J., & Gu, C. (2023). Detection of Android Malware Based on Deep Forest and Feature Enhancement. IEEE Access, 11, 29344-29359. doi: 10.1109/ACCESS.2023.3260977

ii. He, H., Li, X., Chen, P., Chen, J., Liu, M.,... Wu, L. (2024). Efficiently localizing system anomalies for cloud infrastructures: a novel Dynamic Graph Transformer based Parallel Framework. Journal of Cloud Computing, 13(1), 115. doi: https://doi.org/10.1186/s13677-024-00677-x

i. Xuemin, Z., Haitao, D., Zenggang, X., Ying, R., Yanchao, L., Yuan, L.,... Delin, H. (2024). Self-Organizing Key Security Management Algorithm in Socially Aware Networking. Journal of Signal Processing Systems, 96(6), 369-383. doi: https://doi.org/10.1007/s11265-024-01918-7

ii. Wang, G., Yang, J., & Li, R. (2017). Imbalanced SVM-Based Anomaly Detection Algorithm for Imbalanced Training Datasets. ETRI Journal, 39(5), 621-631. doi: https://doi.org/10.4218/etrij.17.0116.0879

iii. Zhang, H., Xu, Y., Luo, R., & Mao, Y. (2023). Fast GNSS acquisition algorithm based on SFFT with high noise immunity. China Communications, 20(5), 70-83. doi: 10.23919/JCC.2023.00.006

iv. Li, X., Lu, Z., Yuan, M., Liu, W., Wang, F., Yu, Y.,... Liu, P. (2024). Tradeoff of Code Estimation Error Rate and Terminal Gain in SCER Attack. IEEE Transactions on Instrumentation and Measurement, 73, 1-12. doi: 10.1109/TIM.2024.3406807

v. Zhang, J., Yang, D., Li, W., Zhang, H., Li, G.,... Gu, P. (2024). Resilient Output Control of Multiagent Systems With DoS Attacks and Actuator Faults: Fully Distributed Event-Triggered Approach. IEEE Transactions on Cybernetics, 1-10. doi: 10.1109/TCYB.2024.3404010

vi. Zhou, P., Peng, R., Xu, M., Wu, V., & Navarro-Alarcon, D. (2021). Path Planning With Automatic Seam Extraction Over Point Cloud Models for Robotic Arc Welding. IEEE Robotics and Automation Letters, 6(3), 5002-5009. doi: 10.1109/LRA.2021.3070828

2- The criteria for selecting the 43 items in the catalog are not clearly explained. Providing a detailed justification for the selection of each item, including references to specific studies or industry standards, would improve clarity.

3- The survey used to evaluate the catalog is not described in enough detail, particularly regarding participant selection and survey design. A more comprehensive description of the survey methodology, including participant selection, survey design, and response analysis, is recommended.

4- The paper does not adequately explain the validation process for the catalog items. Elaborating on the validation process, including specific steps taken to ensure the reliability and validity of the catalog items, would enhance credibility.

5- The paper acknowledges the subjectivity of collaboration but does not propose concrete methods to address this issue in requirements elicitation. Suggesting specific techniques or tools to mitigate subjectivity in the elicitation of collaboration requirements would be beneficial.

6- The paper does not provide practical examples or case studies demonstrating the application of the catalog. Including case studies or examples of real-world applications of the catalog would support its practical relevance.

7- There is no discussion on how the proposed catalog integrates with existing requirements engineering frameworks or methodologies. Discussing the integration of the catalog with current requirements engineering practices and frameworks would aid practitioners in its implementation.

8- The focus on non-functional requirements might limit the scope of the catalog. Including functional requirements relevant to collaborative software would provide a more holistic approach.

9- The technical aspects of how collaboration features can be implemented are not sufficiently covered. Adding more technical details on the implementation of collaboration features, including possible technologies and architectures, would add depth.

10- The paper’s future work section is brief and does not address scalability issues. Expanding the future work section to explore scalability in different organizational contexts and larger projects, including potential challenges and solutions, would provide a more comprehensive outlook.

6. PLOS authors have the option to publish the peer review history of their article (what does this mean?). If published, this will include your full peer review and any attached files.

Reviewer #1: **Yes: **Muhammad Asim

Reviewer #2: **Yes: **Dr. Islam Zada, Department of Software Engineering, International Islamic University Islamabad, Pakistan.

---

## [Author Response · Author response to Decision Letter 1]

16 Jul 2024

Respected Professor,

Greetings of the day.

I have revised the manuscript as per your guidelines & improved it.

Hope to hear from you soon.

Thanks

---

## [Decision Letter · Decision Letter 1]

15 Aug 2024

An Efficient Detection of Sinkhole Attacks using Machine Learning: Impact on Energy and Security

PONE-D-24-17513R1

Dear Dr. Muhammad Zulkifi Hasan,

We’re pleased to inform you that your manuscript has been judged scientifically suitable for publication and will be formally accepted for publication once it meets all outstanding technical requirements.

Kind regards,

Gauhar Ali, Ph.D

Academic Editor

PLOS ONE

Additional Editor Comments (optional):

Reviewers' comments:

Reviewer's Responses to Questions

**Comments to the Author**

1. If the authors have adequately addressed your comments raised in a previous round of review and you feel that this manuscript is now acceptable for publication, you may indicate that here to bypass the “Comments to the Author” section, enter your conflict of interest statement in the “Confidential to Editor” section, and submit your "Accept" recommendation.

Reviewer #1: All comments have been addressed

Reviewer #2: All comments have been addressed

2. Is the manuscript technically sound, and do the data support the conclusions?

Reviewer #1: Yes

Reviewer #2: Yes

3. Has the statistical analysis been performed appropriately and rigorously? 

Reviewer #1: Yes

Reviewer #2: Yes

4. Have the authors made all data underlying the findings in their manuscript fully available?

Reviewer #1: Yes

Reviewer #2: Yes

5. Is the manuscript presented in an intelligible fashion and written in standard English?

Reviewer #1: Yes

Reviewer #2: Yes

6. Review Comments to the Author

Reviewer #1: (No Response)

Reviewer #2: My Recommendations are properly incorporated. I recommended for acceptance.

My Recommendations are properly incorporated. I recommended for acceptance.

My Recommendations are properly incorporated. I recommended for acceptance.

My Recommendations are properly incorporated. I recommended for acceptance.

7. PLOS authors have the option to publish the peer review history of their article (what does this mean?). If published, this will include your full peer review and any attached files.

Reviewer #1: No

Reviewer #2: **Yes: **Dr.Islam Zada

---

## [Editor Report · Acceptance letter]

PONE-D-24-17513R1

PLOS ONE

Dear Dr. Hasan,

I'm pleased to inform you that your manuscript has been deemed suitable for publication in PLOS ONE. Congratulations! Your manuscript is now being handed over to our production team.

Kind regards,

on behalf of

Dr. Gauhar Ali

Academic Editor

PLOS ONE